



# Trends and Emissions of Six Perfluorocarbons in the Northern and Southern Hemisphere

Elise S. Droste[1], Karina E. Adcock[1], Matthew J. Ashfold[2], Charles Chou[3], Zoë Fleming[4,*], Paul J. Fraser[5], Lauren J. Gooch[1], Andrew J. Hind[1], Ray L. Langenfelds[5], Emma Leedham Elvidge[1], Norfazrin Mohd Hanif[1,6], Simon O'Doherty[7], David E. Oram[1,8], Chang-Feng Ou-Yang[9], Marios Panagi[4], Claire E. Reeves[1], William T. Sturges[1], and Johannes C. Laube[1,10]

[1]Centre for Ocean and Atmospheric Sciences, School of Environmental Sciences, University of East Anglia, Norwich, NR4 7TJ, UK
[2]School of Environmental and Geographical Sciences, University of Nottingham Malaysia, 43500 Semenyih, Malaysia
[3]Research Center for Environmental Changes, Academia Sinica, Taipei 11529, Taiwan
[4]National Centre for Atmospheric Science (NCAS), Department of Chemistry, University of Leicester, UK
[5]Commonwealth Scientific and Industrial Research Organisation, Oceans and Atmosphere, Climate Science Centre, Aspendale, Australia
[6]School of Environmental and Natural Resource Sciences, Faculty of Science and Technology, Universiti Kebangsaan Malaysia, 43600 Bangi, Selangor, Malaysia
[7]Department of Chemistry, University of Bristol, Bristol, UK
[8]National Centre for Atmospheric Science, School of Environmental Sciences, University of East Anglia, Norwich, NR4 7TJ, UK
[9]Department of Atmospheric Sciences, National Central University, Taoyuan, Taiwan
[10]Institute of Energy and Climate Research – Stratosphere (IEK-7), Forschungszentrum Jü lich GmbHJ, Jülich, Germany
*now at Center for Climate and Resilience Research (CR2), University of Chile, Santiago, Chile

**Correspondence:** E. S. Droste (e.droste@uea.ac.uk)

**Abstract.** Perfluorocarbons (PFCs) are potent greenhouse gases with Global Warming Potentials up to several thousand times greater than $CO_2$ on a 100-year time horizon. The lack of any significant sinks for PFCs means that they have long atmospheric lifetimes on the order of thousands of years. Anthropogenic production is thought to be the only source for most PFCs. Here we report an update on the global atmospheric abundances of the following PFCs, most of which have for the first time been

5     separated according to their isomers: c-octafluorobutane (c-$C_4F_8$), n-decafluorobutane (n-$C_4F_{10}$), n-dodecafluoropentane (n-$C_5F_{12}$), n-tetradecafluorohexane (n-$C_6F_{14}$), and n-hexadecafluoroheptane (n-$C_7F_{16}$). Additionally, we report the first data set on the atmospheric mixing ratios of perfluoro(2-methylpentane) (i-$C_6F_{14}$). The existence and significance of PFC isomers has not been reported before, due to the analytical challenges of separating them. The time series spans a period from 1978 to the present. Several datasets are used to investigate temporal and spatial trends of these PFCs: time series of air samples

10     collected at Cape Grim, Australia, from 1978 to the start of 2018; a time series of air samples collected between July 2015 and April 2017 at Tacolneston, UK; and intensive campaign-based sampling collections from Taiwan. Although the remote "background" Southern Hemispheric Cape Grim time series indicates that recent growth rates of most of these PFCs are lower than in the 1990s, we continue to see significantly increasing mixing ratios that are between 6 % to 27 % higher by the end of 2017 compared to abundances measured in 2010. Air samples from Tacolneston show a positive offset in PFC mixing ratios



compared to the Southern Hemisphere baseline. The highest mixing ratios and variability are seen in air samples from Taiwan, which is therefore likely situated much closer to PFC sources, confirming predominantly Northern Hemispheric emissions for most PFCs. Even though these PFCs occur in the atmosphere at levels of parts per trillion molar or less, their total cumulative global emissions translate into 833 million metric tonnes of $CO_2$ equivalent by the end of 2017, 23 % of which has been

5 emitted in the last eight years. Almost two-thirds of the $CO_2$ equivalent emissions are attributable to c-$C_4F_8$, which currently also has the highest emission rates that continue to grow. Despite this, the sources of all PFCs covered in this work remain poorly constrained and reported emissions in global databases do not account for the abundances found in the atmosphere.



# 1 Introduction

Perfluorocarbons (PFCs) are fully fluorinated hydrocarbon chemicals that occur ubiquitously in the atmosphere. Even though the magnitude of their mixing ratios are on the order of parts per trillion molar (ppt) or less, monitoring their atmospheric concentrations is highly warranted by their strong infra-red absorption properties and inertness. As a result, PFCs can have global

warming potentials (GWPs) of several thousands on a 100-year time horizon and extremely long atmospheric lifetimes (Bravo et al., 2010; Hartmann et al., 2013) (Table 1). Even though their low abundances mean that their contribution to radiative forcing is relatively minor at present, their continued accumulation in the atmosphere might result in a significant contribution to climate change by the end of the century (Zhang et al., 2011). Previous studies have shown the emergence of atmospheric PFCs in the 1960s based on firn air samples and a subsequent accelerated increase in the 1990s (Worton et al., 2007; Mühle et al.,

2010; Ivy et al., 2012a; Laube et al., 2012; Oram et al., 2012; Trudinger et al., 2016). With the exception of tetrafluoromethane ($CF_4$) (Harnisch et al., 1996; Deeds et al., 2008; Mühle et al., 2010), all known PFCs are thought to be almost exclusively anthropogenic in origin (EDGAR, 2014). For the lowest molecular weight PFCs, such as $CF_4$ and hexafluoroethane ($C_2F_6$), emissions from aluminium production are one of the largest sources. Other sources include the use in chemical vapour deposition (CVD) chamber cleaning, plasma etching, refrigerants, fire protection, fluoropolymer production, and - especially for the

higher molecular weight PFCs, such as n-$C_6F_{14}$ and n-$C_7F_{16}$ - use as heat transfer fluids and solvents (Robin and Iikubo, 1992; Mazurin et al., 1994; Beu, 2005; EPA, 2008; Hartmann et al., 2013; EDGAR, 2014; Mühle et al., 2019). Atmospheric trends vary, but mixing ratios are increasing for all known PFCs. Previously reported PFC trends in the atmosphere include $CF_4$, $C_2F_6$, octafluoropropane ($C_3F_8$), cyclic-octafluorobutane (c-$C_4F_8$), decofluorobutane ($C_4F_{10}$,) dodecafluoropentane ($C_5F_{12}$), tetradecafluorohexane ($C_6F_{14}$), hexodecafluoroheptane ($C_7F_{16}$), and octadecafluorooctane ($C_8F_{18}$) (Laube et al., 2012; Ivy

et al., 2012a; Oram et al., 2012; Mühle et al., 2010; Mühle et al., 2019).

The current work focuses on six PFC isomers: c-$C_4F_8$, n-$C_4F_{10}$, n-$C_5F_{12}$, n-$C_6F_{14}$, i-$C_6F_{14}$ (perfluoro-2-methylpentane), and n-$C_7F_{16}$. Even though all but one of these PFCs have been reported before, this is the first time that we report on their separated isomers, except for c-$C_4F_8$ and n-$C_5F_{12}$, for which we are updating and expanding our previously reported data set (Laube et al., 2012; Oram et al., 2012). The GWPs on a 100-year time horizon of these compounds range between 7820 (n-

$C_7F_{16}$) and 9540 (n-$C_4F_{10}$) (Table 1) (Bravo et al., 2010; Hartmann et al., 2013). To assess the recent changes in atmospheric abundances of these PFCs, this work aims to achieve the following objectives:

1. To extend the 1978-2010 PFC records from Cape Grim, Australia (Oram et al., 2012; Laube et al., 2012) into early 2018. The air sampled at Cape Grim under baseline conditions has typically travelled over the open ocean many days prior and is known to be very well-mixed (Fraser et al., 1999). The Cape Grim time series is therefore an important data set

used to study Southern Hemisphere baseline trends of various trace gases (Oram et al., 1995; Langenfelds et al., 1996; Fraser et al., 1999; Reeves et al., 2005; Laube et al., 2012; Adcock et al., 2018). The global trend between 1978 and 2010 for c-$C_4F_8$ has been discussed in Oram et al. (2012) and most recently in Mühle et al. (2019), while those for n-$C_4F_{10}$, n-$C_5F_{12}$, n-$C_6F_{14}$, and n-$C_7F_{16}$ have previously been investigated in Laube et al. (2012) and Ivy et al. (2012a). Data acquired since then allows for a re-analysis of the conclusions, given the most recent trends.




2. To transfer some measurements to a new calibration scale. Re-calibration of the PFCs was necessary, as a number of PFC isomers have recently been identified in our laboratory at the University of East Anglia (UEA), including two isomers of $C_4F_{10}$ (n-$CF_3(CF_2)_2CF_3$ and i-$CF(CF_3)_3$), two isomers of $C_6F_{14}$ (n-$CF_3(CF_2)_4CF_3$ and i-$CF_3(CF_2)_2CF(CF_3)_2$), and three isomers of $C_7F_{16}$, of which one is the n-isomer ($CF_3(CF_2)_5CF_3$). The chemical structure of the other two isomers is still unknown. A suspicion that the chromatogram peaks previously reported for $C_6F_{14}$ and $C_7F_{16}$ consisted of more than one isomer had already been expressed by Laube et al. (2012). Since then, a new gas-chromatography method reported here (Section 2.2) has been developed that led to the separation of these isomers, consequently increasing the accuracy of the measurements.

3. To study the atmospheric trend and abundance of i-$C_6F_{14}$ for the first time. The separation of isomers allows for more accurate estimations of the impact on the atmospheric radiative forcing, especially since i-$C_6F_{14}$ has a different radiative efficiency than n-$C_6F_{14}$ (Bravo et al., 2010).

4. To compare the abundance and variability of PFCs between the well-mixed, unpolluted air in the Southern Hemisphere, which are well represented by the Cape Grim data, and the Northern Hemisphere, where most of the emissions occur. Regarding the latter, this study presents recent observations from air samples collected in Tacolneston, United Kingdom, and in Taiwan. Neither of these sites necessarily represent average abundances for the Northern Hemisphere, but the variability in the observations gives information on the geographical distance of these sites to potential sources. This leads to the final objective:

5. To estimate the global emissions of these six PFCs and compare them to emissions reported in the EDGARv4.2 database (EDGAR, 2014). PFCs are a part of the 1997 Kyoto Protocol and are covered in the 2014-revised F-Gas regulation of the European Union (EU), which will be a part of the Nationally Determined Contribution of the EU to the Paris Agreement. However, substantial discrepancies have been found between emissions derived from atmospheric background measurements and reported emissions (specifically for the compounds focused on in this study, see Laube et al. (2012), Oram et al. (2012), Ivy et al. (2012b), Mühle et al. (2019)). The gas-chromatographic separation of isomers is a useful development that may give some insight into, for example, the identity and potential commonality of sources and possibly a better understanding of the changes of sources with time.

The methods are discussed in Section 2, including sampling, analysis, emission modelling, uncertainties, and air mass trajectory modelling. The results are split up into three sections. The first section (Section 3.1) presents the Cape Grim observations and trends since 1978 for each PFC, with a focus on data since 2010. These are then compared to measurements from Tacolneston and Taiwan. The second section (Section 3.2) discusses the emissions based on the emission modelling and compares these to the emissions from the emission report-based EDGARv4.2 database. The third section of the results (Section 3.3) covers potential source regions and types based on air mass trajectories simulated by the NAME model for the Taiwan samples. Implications of the findings are outlined in Section 4. Section 5 summarises the main conclusions.





## 2 Methods

### 2.1 Sampling and Measurement Sites

The surface air samples from Cape Grim, Tacolneston, and two sites in Taiwan have been collected over different periods of time and at different frequencies (Table 2).

5    1. Air samples collected at Cape Grim Baseline Air Pollution Station, Tasmania, since 1978 have been analysed regularly at UEA since the early 1990s. Between 1978 and 1994 and again since 2012, samples were collected as sub-samples from parent air archive tanks. Between 1995 and 2011, the majority of samples were collected directly in UEA flasks (3 L) and only six samples were sub-sampled from parent air archives. Sampling procedures for this period are outlined in Langenfelds et al. (1996), Oram et al. (1995), and Fraser et al. (1999), and confidence in the stability of the PFCs in the archive is justified in Oram et al. (2012). A selection of these samples have been re-analysed using an updated method in order to revise the n-$C_4F_{10}$, n-$C_6F_{14}$, and n-$C_7F_{16}$ mixing ratios previously published in Laube et al. (2012) (see Section 2.2). Samples for UEA have been collected since 1994 in electropolished stainless steel canisters (Rasmussen) until 2000 and in silcosteel-treated stainless steel canisters (Restek Corp.) since 2001 (Laube et al., 2012; Oram et al., 2012). Air samples collected for the Cape Grim Air Archive and UEA represent well-mixed, clean, Southern Hemispheric air, as the sampled air masses originate from the long trajectories over the Southern Ocean. Hence, the trace gas abundances in these samples can be considered as mid-latitude Southern Hemispheric background levels.

    2. Samples from Tacolneston, United Kingdom, have been collected on a near bi-weekly basis between July 2015 and April 2017 (Stanley et al., 2018; Adcock et al., 2018). Samples were collected in Silcosteel-treated stainless steel canisters (Restek Corp.).

20    3. In Taiwan, air samples were collected between 2013 and 2016, in March and April each year. The collection site alternated each year between a site in the north (Cape Fuguei) and in the south of Taiwan (Hengchun), starting in the south in 2013 (Laube et al., 2016; Oram et al., 2017; Adcock et al., 2018). Sample collection was aimed at the winter monsoon flow from the north. Even though Hengchun is in the south and air at this time of year travels primarily from the north over Taiwan, the air at Hengchun has a similar trace gas composition as air in Cape Fuguei. This is because Taiwan's topography re-directs the northern winds along the eastern side of the mountains where the population is sparse and thus does not pick up additional anthropogenic PFC emissions before it arrives at Hengchun. Due to local PFC contamination issues during sampling, measurements taken in 2013 have been excluded.

### 2.2 Analytical Methods

All air samples were analysed using an Agilent 6890 Gas Chromatograph coupled to a Waters AutoSpec magnetic sector mass spectrometer (AutoSpec GC-MS). An in-house built cryo-trapping system pre-concentrates 200-300 mL of the air sample at -78 °C in a HayeSep D-filled loop (80/100 mesh) after passing through a magnesium perchlorate ($Mg(ClO_4)_2$) drier to remove



any water vapour. Two columns are routinely used as part of AutoSpec analysis at UEA: a GS-GasPro column (length ~30-50 m; ID: 0.32 mm) and a KCl-passivated CP-PLOT $Al_2O_3$ column (length: 50 m; ID: 0.32 mm). As many samples as possible are analysed on both columns to provide a large suite (50+ compounds) of trace gas measurements (e.g. Adcock et al., 2018; Laube et al., 2016; Leedham Elvidge et al., 2018) (Table 2). For the current work, the CP-PLOT column is the most relevant,

as it has the ability to separate the higher molecular weight PFCs and their isomers based on their boiling point and symmetry. For compounds analysed on the CP-PLOT column, an Ascarite (NaOH-coated silica) trap is included before the magnesium perchlorate trap to remove $CO_2$, which can distort or reduce the signal of a number of compounds. The sample is injected into the GC at 100 °C. Instrumental drift was accounted for by bracketing samples with calibrated working standards (see Section 2.3).

The Taiwan samples collected in 2014 were analysed on the GS-GasPro column only, using a method optimised for a particular set of trace gases that does not include isomers of $C_6F_{14}$ and $C_7F_{16}$. Taiwan samples collected in 2015 and 2016, as well as all Tacolneston samples, are measured on the CP-PLOT column.

The dry air mole fractions are measured and the mixing ratios (in units of parts per trillion, ppt) reported in this study are used as an equivalent to picomole per mole. The absolute calibration method has been described in detail in Laube et al. (2010)

and has been described specifically in Oram et al. (2012) for c-$C_4F_8$ and in Laube et al. (2012) for $C_4F_{10}$, $C_5F_{12}$, $C_6F_{14}$ and $C_7F_{16}$, including linearity of the detector response and identification. The ions ($m/z$) used for the quantification of c-$C_4F_8$, n-$C_4F_{10}$, and n-$C_5F_{12}$ are: 131.0, 119.0, and 119.0, respectively. i-$C_6F_{14}$, n-$C_6F_{14}$, and n-$C_7F_{16}$ have been analysed on both quantifying ions ($m/z$) 169.0 and 219.0. Ion $m/z$ 169 is used for trend analysis, unless a baseline distortion occurred in the chromatogram, in which case the 219 ion is used. Other studies that used the same set-up and method include Laube et al.

(2016) and Adcock et al. (2018).

Even though two isomers of $C_4F_{10}$, two isomers of $C_6F_{14}$, and three isomers of $C_7F_{16}$ have been identified with the current analytical method, the current study only focuses on a selection of these (n-$C_4F_{10}$, i-$C_6F_{14}$, n-$C_6F_{14}$, n-$C_7F_{16}$) for the following reasons. The main signal of the quantifying ion for i-$C_4F_{10}$ ($m/z$ 131.0) is not well separated from the larger n-$C_4F_{10}$ peak on that same ion. Nevertheless, it is still possible to determine the n-isomer for $C_4F_{10}$ on a different quantifying ion, which we

confirm to be unaffected by the i-isomer (see Section 2.3). Regarding the isomers of $C_7F_{16}$, two out of the three isomers have very small signals resulting in bad precisions. Additionally, these two smallest peaks are not well separated from each other and the number of possible isomers with very similar mass spectra is too high to allow for a high-confidence identification. Therefore, their current quantification is too inaccurate and imprecise. However, it is possible to quantify the n-isomer of $C_7F_{16}$ with sufficient precision, because it has a larger signal. For the same reasons, both isomers of $C_6F_{14}$ are quantified here.

**2.3   Calibrations**

The previously reported calibration scales (Laube et al., 2012) were revised to accommodate the separated isomers. The calibration procedure has been described in Laube et al. (2010) and has undergone little alteration. High purity compounds for n-$C_4F_{10}$, n-$C_6F_{14}$, i-$C_6F_{14}$, and n-$C_7F_{16}$ were diluted to ppt levels in a two-step static dilution series using Oxygen Free Nitrogen (OFN) gas (British Oxygen Company). All pure compounds had a 98 % purity or higher at time of acquisition and





were subsequently further purified in our lab by subjecting each to three repeated freeze-heating cycles. The OFN was filtered to remove small amounts of trace gas contaminations by flowing the gas through 60 cm $\frac{1}{4}$" stainless-steel tubing filled with HayeSep D, which was submersed in an ethanol-dry-ice cold trap (-78 °C). The calibration system operates under low pressures (<300 mbar) and high temperatures (100 °C). This results in insignificant virial coefficients, and thus the dry-air mole

fractions could be calculated using the ideal gas law (Laube et al., 2010).

n-$C_4F_{10}$ was re-calibrated to rule out any bias from the observed i-$C_4F_{10}$ isomer, and to confirm the leak-tightness of the calibration system (Laube et al., 2016). The combined influence of the i-isomer and the leak-tightness of the calibration system is <2.8 %. This has been determined by comparing the old and the new calibration scale for n-$C_4F_{10}$. Due to the leak-tightness of the system and a lack of observed isomers, a revision of the calibration scales for c-$C_4F_8$ and n-$C_5F_{12}$ was

deemed unnecessary. Note that the data for c-$C_4F_8$, n-$C_4F_{10}$, and n-$C_5F_{12}$ in Oram et al. (2012) and Laube et al. (2012) are included in the current work.

The confidence in the complete separation of all isomers for $C_6F_{14}$ and $C_7F_{16}$ is based on the ratio of the two most abundant ions into which these compounds are ionised during mass spectrometry: (mass-to-charge (*m/z*) ratios 219.0 and 169.0). A comparison of the 219:169 ion ratio is made between the Cape Grim air samples and the calibrations done (Fig. S2). Since

the calibration samples are based on dilutions of high purity isomer compounds (>98 %), a significant deviation from the ion ratios in the air samples compared to the ion ratios in the calibration samples might suggest that the peak measured in the chromatogram for the air samples actually consists of more than one isomer. In turn, if the ion ratios measured in the air samples are similar to the ion ratios measured in the calibration samples, then confidence can be attributed within the uncertainties that the signal measured in the air samples is for one particular isomer only. For i-$C_6F_{14}$, n-$C_6F_{14}$, and n-$C_7F_{16}$, the ion ratio in

the Cape Grim samples did not differ significantly from the ion ratio in the calibrations (Fig. S2). It should be noted that even if other isomers do exist and co-elute, they a) likely do not have a trend that is substantially different from any of the $C_6F_{14}$ and $C_7F_{16}$ isomers discussed here and b) currently have mixing ratios that are extremely small compared to those of the main isomers.

Calibrated mixing ratios for the working standard (i.e. unpolluted Northern Hemispheric air from 2017) are given in Table

4, along with the average analytical precision per compound. To determine the accuracy of the calibrations, trichlorofluoromethane (CFC-11) was diluted along with the pure PFCs as a reference compound, as our working tank has been calibrated by the globally recognised CMD (Global Measurement Division) of the NOAA-ESRL (National Oceanic and Atmospheric Administration - Earth System Research Laboratory) for CFC-11. For all UEA calibrations, the average difference with values determined by NOAA show that the UEA calibrated concentrations for CFC-11 are consistently slightly lower by on average

4.2 % ± 0.3 %. To put the accuracy of these calibrations into perspective, the accuracies of calibrations for the PFCs in common with Ivy et al. (2012a) and Laube et al. (2012) are between 4.0 % and 7.8 % and between -5.5 % and 2.8 %, respectively.

The improvement of the new calibration scale by separating the isomers of $C_6F_{14}$ and $C_7F_{16}$ is substantial. Even though the n-isomer of both $C_6F_{14}$ and $C_7F_{16}$ is dominant for both PFCs, the old calibration scale would have overestimated the mixing ratios of these n-isomers by 20 % and 11 %, respectively. The new UEA calibration scale thus allows for a more accurate

analysis of atmospheric mixing ratios of these PFCs.



## 2.4 Emission Modelling

The annual global emissions are derived using a 2-D global atmospheric chemistry-transport model (run using the software Facsimile, version 7, also see Laube et al. (2012); Oram et al. (2012); Reeves et al. (2005); Fraser et al. (1999)). The model domain consists of 24 equal-area, zonally averaged latitudinal bands, which have twelve horizontal layers. Each layer is attributed

with a height of 2 km, resulting in a total altitude of 24 km.

Due to the high stability of the C-F bonds, PFCs are considered to be chemically inert and thus the model is set to have no chemical or photolytic loss for these compounds. Photolytic loss would only be relevant at stratospheric and mesospheric altitudes (Morris et al., 1995), which are not fully represented in the model domain. The only sink for PFCs in the model is the diffusive loss at the top boundary of the model domain, where the diffusive loss is controlled by a fixed ratio of the mixing

ratio of the PFC in the top atmospheric layer to the layer above the model domain, thereby creating a gradient. If there is no gradient, i.e. the ratio is equal to 1, then there is no diffusive loss. The upper boundary ratio used for each modelled gas is set such that the diffusive loss replicates the lifetime of the gas in the upper atmosphere (i.e. above the model domain). For the PFCs, this is effectively their atmospheric lifetime. As PFCs have lifetimes that are much longer than the time period studied, the gradient out of the model is set very low (i.e. ratio of 0.997; Laube et al. (2012)).

The emission distribution is based on the global distribution of reported $C_6F_{14}$ emissions in the EDGARv4.2 data set in 2005. This set-up is kept the same as in Laube et al. (2012), because the EDGARv4.2 emission distribution for $C_6F_{14}$ between 2005 and 2010 has not changed significantly and also does not significantly affect the simulated mixing ratios. The reason the EDGARv4.2 data for $C_6F_{14}$ is used for all PFCs is that the discrepancy between atmospherically-derived and reported emissions is smallest for $C_6F_{14}$ (Laube et al., 2012). In the model, PFC emissions are set to occur 99 % in the northern mid-

latitudes. This is a realistic assumption according to reported emissions recorded in the EDGARv4.2 database, which indicate that 98 % to 100 % of the PFC emissions occur in the Northern Hemisphere, depending on the PFC compound (EDGAR, 2014).

Model runs start in 1934 and end in 2018. The annual emissions are iteratively altered to obtain a best fit of the modelled mixing ratios to the observed mixing ratios at Cape Grim (located within the latitudinal band spanning 35.7 °S to 41.8 °S).

The earliest annual PFC emissions set in the model start in the 1950s, when certain PFCs were first detected above detection limits in firn air (Laube et al., 2012). The best fit is determined by minimising the sum of least squares between the modelled and observed Cape Grim mixing ratios (Table S2).

## 2.5 Uncertainties

Two versions of uncertainties have been computed for the model-derived PFC trends for Cape Grim and Tacolneston data

(visualised as the shaded envelopes in the figures). The first version will be referred to as the "trend uncertainty", which consists of the annually averaged analytical uncertainty, the averaged model-fit uncertainty of the Cape Grim trend, and a 5 % modelling uncertainty (Table S1). The modelling uncertainty accounts for uncertainties regarding the model transport scheme and is the error in simulating the concentration at Cape Grim of a long-lived gas that is emitted primarily in the Northern





Hemisphere at a well known rate (Reeves et al., 2005). Lifetime uncertainties are not included, because they are too long for the time period studied here and would not have a significant effect on the results. Note that the trend uncertainty does not include the calibration uncertainty. Even though it is important to consider the calibration uncertainty in order to evaluate the complete uncertainty of the model simulation, it does not affect the trend. However, to give an overview of all uncertainties, the calibration is included in the "total uncertainty", which is the second type of uncertainty referred to in this work (Table S1). An additional uncertainty for the trends of n-$C_4F_{10}$ and n-$C_5F_{12}$ is added in the total uncertainty, which is the error in the conversion of the mixing ratio between internal working standards, as these two PFCs have not been re-calibrated for the current study (see Section 2.3).

After these uncertainties were calculated, the model was re-run using the best fit of the observed mixing ratios adjusted by the uncertainties to estimate the maximum and minimum emissions. Measurement errors (indicated by the error bars on the observational data points) consist of a combination of the $1\sigma$ standard deviations of the working standard and sample replicates on the same analysis day. For samples that have been analysed against the previous working standard, the uncertainties include the aforementioned internal conversion accounted for as the $1\sigma$ standard deviation of the peak ratio on inter-comparison days (n=8).

## 2.6 NAME Modelling

This work used the UK Met Office's Langrangian particle dispersion model, Numerical Atmospheric Modelling Environment (NAME) (Jones et al., 2007), for tracking and understanding the origin of air masses arriving at Taiwan. The model was run in the backward mode for 12-day long simulations to generate the footprints of where the air sampled during the campaigns had previously been close to the Earth's surface (see Fig. 1). The analysis begins by releasing batches of 30 000 inert backward particles over a three hour period encompassing the collection time of each sample. Over the course of the 12-day travel time, the locations of all particles within the lowest 100 m of the model atmosphere were recorded every 15 minutes on a grid with a resolution of 0.25° longitude and 0.25° latitude. The trajectories were calculated using three-dimensional meteorological fields produced by the UK Met Office's Numerical Weather Prediction tool, the Unified Model (UM). These fields have a horizontal grid resolution of 0.23° longitude by 0.16° latitude and 59 vertical levels below ∼30 km.

In order to quantify the contribution of various regions to each footprint, the domain was divided into 15 regions using shape files produced by ArcGIS, a geographic information system (GIS) for working with maps and geographic information (Fig. 2). The segregation of China from the East Asia region category has enabled a more detailed analyses to be conducted to determine which specific Chinese sub-regions could contribute to the variations in the PFCs mixing ratios in Taiwan. The contribution of each region is quantified by summing the particle concentration ($gsm^{-3}$) in each grid cell within each shape file (see Fleming et al. (2012) and O'Shea et al. (2017)).

For the purpose of identifying the possible sources of PFCs, carbon monoxide (CO) emission data were used along with the NAME footprints to calculate a modelled mixing ratio of CO at the Taiwan measurement site. This modelled CO only takes into account the CO that has been emitted during the 12-day simulation in the NAME model domain (more details can be found in Oram et al. (2017). CO was used, because it is a tracer of anthropogenic emissions and its lifetime of 1–2 months is long enough





to track pollution plumes on the regional scale of this study. Additionally, reasonable bottom-up emissions of CO also exist and are widely used and tested. The inventory of CO emissions was taken from the Representative Concentration Pathway 8.5 (RCP 8.5) (Riahi et al., 2011; Van Vuuren et al., 2011) for the year 2010. More information and access to the database can be found on the RCP 8.5 website (http://tntcat.iiasa.ac.at:8787/RcpDb/dsd?Action=htmlpage&page=welcome). The CO emission

inventory is divided into various emission sectors: industry, power plants, solvents, agricultural waste burning, waste, forest burning, grassland burning, residential, international shipping, surface transportation, and agriculture. A correlation analysis was performed between the modelled CO and the measured PFC mixing ratios in air samples collected in Taiwan in order to assess the extent to which these categories of emissions might be related to the PFC mixing ratios in these air masses. Industry (combustion and processing) and solvent applications are expected to show some correlations with PFC concentrations, as they

are the most closely associated with PFC sources. This approach sheds some light on the potential sources of PFCs. Results of these analyses are found in Section 3.3.

## 3  Results and Discussion

### 3.1  Atmospheric abundances

#### 3.1.1  c-$C_4F_8$ Trends

Out of all PFC compounds reported in this work, c-$C_4F_8$ is the most abundant. c-$C_4F_8$ seems to still have substantial Northern Hemispheric emissions, for which the measured abundances provide three potential sources of evidence: 1) a considerable, and even accelerating, rate of increase in concentration over time, 2) the existence of a relatively large interhemispheric gradient given its long atmospheric lifetime, and 3) large variations above background levels at Northern Hemispheric sites. Each of these sources of evidence will be illustrated with the Cape Grim, Tacolneston, and Taiwan observations, respectively.

The c-$C_4F_8$ mixing ratio in the Southern Hemisphere increased from 0.31 ppt in 1978 to 1.52 ppt by the end of 2017 (Fig. 3). This is slightly less than the mixing ratios of ∼1.6 ppt reported in the recent work of Mühle et al. (2019) with the difference most likely being due to the two independent absolute calibration scales. As reported previously by Oram et al. (2012), the mixing ratios seemed to stabilise in the 1990s, but picked up again in the 2000s. The extended time series since 2008 indicate a continuing increasing trend similar to the one pre-1990, totalling in a 27 % enhancement in the last eight years (2010-2018).

In fact, atmospheric c-$C_4F_8$ abundances show accelerating growth in recent years, potential reasons for which are explored in Section 3.2.1.

The mixing ratios at Tacolneston, UK, seem to have a consistent offset compared to the observations at Cape Grim on average of 0.07 ppt (based on monthly averages; Fig. 3), equivalent to an interhemispheric ratio of 1.05. Note that mixing ratios at Tacolneston have not been shown to be representative of Northern Hemisphere background mixing ratios. However,

the lack of variability in the data suggests that Tacolneston is not in close proximity to any major sources of c-$C_4F_8$. When considering the simulated atmospheric trend from the emission model for Tacolneston, it can be noted that it overestimates the observed concentrations, despite the well-fitted Southern Hemispheric trend. The difficulty in validating trace gas trends





for Tacolneston is three-fold. 1) The time series for Tacolneston only covers the time period between 2015 and 2017, which does not enable a validation of the atmospheric trends over the period during which c-$C_4F_8$ has been emitted. 2) The model is 2-dimensional and as such represents zonal averages. In reality, emissions will occur at specific longitudes, which leads to heterogeneity in atmospheric concentrations at the latitudes of the emissions, even for these very long-lived compounds.

This is clearly illustrated by the Taiwan data (see below). 3) The mixing ratios simulated by the model are dependent on a given global distribution of emissions, which harbours a substantial amount of uncertainty. Varying the distribution of the 99 % of global emissions within the Northern Hemisphere does not significantly affect the simulated mixing ratios for Cape Grim, but naturally it does for Northern Hemispheric sites, such as Tacolneston. It is likely that the distribution for c-$C_4F_8$ in the Northern Hemisphere requires revision to include, for example, a distribution that changes over time rather than a fixed

distribution (e.g. see a similar discussion for CFC-113a in Adcock et al., 2018). It is not unrealistic to consider a Northern Hemispheric distribution that moves from higher to lower latitudes over time, as the industries that utilise PFCs and other halocarbons, such as the electronics industry, have moved from more northern latitudes to regions such as South-East Asia (Montzka et al. (2009) for hydrochlorofluorocarbons). However, available information on regions that are sources of c-$C_4F_8$ (and other PFCs in general) provide insufficient evidence to base a well-founded emission distribution on (see Section 3.2). The

global emission distribution for c-$C_4F_8$ is thus kept consistent with that for the other PFCs discussed in this work, for which the simulated Tacolneston trend compares reasonably well with the observations, i.e. within the measurement and modelling uncertainties.

    The c-$C_4F_8$ mixing ratios observed in samples collected in Taiwan are contrasting to those at Cape Grim and Tacolneston because of their large day-to-day variability (Fig. 3 B). The maximum measured mixing ratios for each year in Taiwan are on

average 17 % higher than the average simulated mixing ratio in the respective year. This strongly suggests a close proximity to one or more PFC sources. Considering that much of the air sampled in Taiwan during the campaigns has been transported from the direction of China (Laube et al., 2016; Adcock et al., 2018) and that Taiwan itself has a major semi-conductor industry as well (Saito et al., 2010), it is not surprising to find such high levels of c-$C_4F_8$ at these sites.

### 3.1.2    n-$C_4F_{10}$ and n-$C_5F_{12}$ Trends

Between 1978 and 2018, the background mixing ratios for n-$C_4F_{10}$ and n-$C_5F_{12}$ at Cape Grim have increased from 0.02 ppt to 0.18 ppt and from 0.02 ppt to 0.15 ppt, respectively (Fig. 4). In contrast to c-$C_4F_8$, n-$C_4F_{10}$ and n-$C_5F_{12}$ seem to have much smaller Northern Hemispheric sources at the present time. First of all, the Southern Hemispheric mixing ratio growth rate has declined since $\sim$ 2000. n-$C_4F_{10}$ and n-$C_5F_{12}$ mixing ratios at Cape Grim have only increased by 9 % and 6 % between 2010 and 2018, respectively. Even though growth rates are currently slower than they were in the 1990s, the continuing increasing

mixing ratios shown in the extended time series indicate that sources still exist. The similarity in the trends of n-$C_4F_{10}$ and n-$C_5F_{12}$ suggests that these sources might be similar or common. The ratio between n-$C_4F_{10}$ and n-$C_5F_{12}$ has remained relatively stable between 1978 and 2018 at an average of about 1.3±0.07.

    Second of all, as would be expected from a slowly increasing trace gas – and based on the samples collected at Tacolneston and at Cape Grim (Fig. 4) – there does not seem to be an interhemispheric gradient for these PFCs that is discernible given the





measurement uncertainties. This is also captured in the simulated mixing ratios for both sites by the model. If the interhemispheric ratio for n-C$_4$F$_{10}$ and n-C$_5$F$_{12}$ were to be similar to that of c-C$_4$F$_8$, then the differences between the Cape Grim and Tacolneston data would have to be at least 0.01 ppt. This is not the case, but even so, these differences in mixing ratios are too small to discern given the measurement uncertainty.

With the current measurement precisions and low growth rates, the data allow no detailed conclusions on the appropriateness of the global emission distributions used for the model simulations of these two gases. As the Cape Grim observations are representative of the well-mixed Southern Hemispheric air, the similarity in abundances between the Tacolneston and Cape Grim observations might mean that the Tacolneston site is situated relatively far away from n-C$_4$F$_{10}$ and n-C$_5$F$_{12}$ sources. Additionally, it indicates that the emissions of n-C$_4$F$_{10}$ and n-C$_5$F$_{12}$ occur at a low rate, which means that these PFCs will be

relatively well-mixed in the Northern Hemisphere. Note that the n-C$_4$F$_{10}$ data at Tacolneston do display some scatter, but it is well within the uncertainties of the Cape Grim record.

     Third of all, smaller Northern Hemispheric sources of n-C$_4$F$_{10}$ and n-C$_5$F$_{12}$ compared to c-C$_4$F$_8$ are reflected by the smaller elevations of their mixing ratios measured in Taiwan, despite their enhancement compared to Cape Grim measurements (Fig. 4). Maximum values are around 0.25 ppt for n-C$_4$F$_{10}$ and 0.20 ppt for n-C$_5$F$_{12}$. On average, the mixing ratios measured in

Taiwan are 5 % higher than the average mixing ratios simulated in the respective year for Cape Grim for both n-C$_4$F$_{10}$ and n-C$_5$F$_{12}$. Even though this provides some evidence that there are sources of n-C$_4$F$_{10}$ and n-C$_5$F$_{12}$ in East- and South-East Asia, these sources do not appear to have a large impact on the interhemispheric ratio between the unpolluted Northern Hemisphere and Southern Hemisohere sites. Much lower mixing ratio elevations for both n-C$_4$F$_{10}$ and n-C$_5$F$_{12}$ were observed in 2015. This likely related to different air mass origins of samples collected in this particular year (see also Oram et al. (2017); Adcock

et al. (2018)).

### 3.1.3   i-C$_6$F$_{14}$ and n-C$_6$F$_{14}$ Trends

The trends of i-C$_6$F$_{14}$ and n-C$_6$F$_{14}$ are somewhat similar. n-C$_6$F$_{14}$ (CF$_3$(CF$_2$)$_4$CF$_3$) mixing ratios at Cape Grim have increased from 0.01 ppt in 1978 to 0.22 ppt in 2018 (Fig. 5). Its fastest increase in atmospheric abundance occurred in the 1990s and has since slowed down. Mixing ratios appear to be approaching stabilisation, but have still increased by about 9 % since 2010.

Observed long-term trends are similar for i-C$_6$F$_{14}$ (CF$_3$(CF$_2$)$_2$CF(CF$_3$)$_2$), although its atmospheric abundance is much lower (Fig. 5). This points towards commonality or co-location in sources with n-C$_6$F$_{14}$. i-C$_6$F$_{14}$ mixing ratios were <0.01 ppt before 1987, but reached 0.07 ppt by 2018 (note that the earliest observation for i-C$_6$F$_{14}$ shown within the Cape Grim archive is in 1987, due to limited precision of measurements on samples collected prior to that year). In the current work, it is the PFC with the lowest atmospheric abundance throughout the Cape Grim record. Despite the similarities between these two isomers,

a notable difference is observed in their recent rate of change: the relative growth rate of i-C$_6$F$_{14}$ since 2010 ($\sim$ 19 %) is double that of n-C$_6$F$_{14}$. Given their similar physico-chemical properties it is likely that these two isomers are emitted to the atmosphere by the same anthropogenic processes. If their production has not changed much over time, the i-C$_6$F$_{14}$ :n-C$_6$F$_{14}$ ratio is expected to remain relatively stable over that same time period. However, the i-C$_6$F$_{14}$ :n-C$_6$F$_{14}$ ratio has increased





from an average of 0.26±0.01 between 2003 and 2008 to an average of 0.3±0.01 between 2013 and 2018 (Fig. S1). This might indicate a shift in production method or new sources that emit these isomers in different ratios.

Observed Tacolneston mixing ratios for n-$C_6F_{14}$ and i-$C_6F_{14}$ compared to those at Cape Grim show no clear interhemispheric gradient (Fig. 5). The average monthly difference between the observations at these two sites is 0.008 ppt and 0.005 ppt

for n-$C_6F_{14}$ and i-$C_6F_{14}$, respectively, which approaches or even exceeds the differences in mixing ratio one would expect if the interhemispheric ratio was at least that of c-$C_4F_8$ (0.01 ppt for n-$C_6F_{14}$ and 0.003 for i-$C_6F_{14}$). However, such differences in mixing ratios are too small compared to the measurement uncertainties and thus no interhemispheric gradient is discernible based on these data.

The mixing ratios at Tacolneston for the isomers of $C_6F_{14}$ have greater variability in observed mixing ratios compared to

those of the other PFCs discussed in this work. Mixing ratios occasionally exceed the modelled uncertainty envelopes for Tacolneston, such as in 2015. This might imply a plume at the time of sampling. The simulated trend for the Tacolneston data corresponds well with the lower end of the variability in the observations.

No measurements of either n-$C_6F_{14}$ or i-$C_6F_{14}$ were made for Taiwan samples taken in 2014, as these samples were analysed on a GC-column that did not allow for a separation of these isomers (see Section 2.2). Mixing ratios in Taiwan are again

extremely variable on short time scales and reach higher abundances than at Cape Grim or Tacolneston, ranging between 0.21-0.47 ppt for n-$C_6F_{14}$ and between 0.06-0.13 ppt for i-$C_6F_{14}$ (Fig. 5). On average, the mixing ratios measured for both n-$C_6F_{14}$ and i-$C_6F_{14}$ are 9 % higher in Taiwan than at Cape Grim.

Even though a large range of mixing ratios is observed at Tacolneston and in Taiwan for both compounds, the lack of a discernible interhemispheric gradient (given the measurement uncertainties) suggests that the Northern Hemispheric sources

are not as substantial as they are for, for example, c-$C_4F_8$.

### 3.1.4   n-$C_7F_{16}$ Trends

Similarly to c-$C_4F_8$, n-$C_7F_{16}$ shows substantial sources in the Northern Hemisphere, as is suggested by its rate of change, interhemispheric gradient, and large variations above background levels in Taiwan (Fig. 6 A, B). Mixing ratios of n-$C_7F_{16}$ at Cape Grim increased from 0.01 ppt in 1978 to 0.11 ppt in 2018. Its trend is different from those of the other PFC compounds

reported here, as its mixing ratios have been continuously increasing at an approximately constant rate since 1985. Between 2010 and 2018, atmospheric background levels in the Southern Hemisphere have increased by 21 % and show no signs of slowing down. Two statistical outliers appear in measurements on samples collected in 1988 and 1992.

Observations at Tacolneston and Cape Grim together display an interhemispheric gradient for n-$C_7F_{16}$ of 1.04, which is captured by the model. However, even though this interhemispheric gradient is comparable to that for c-$C_4F_8$, the measurement

uncertainties are too large to be able to conclude based on these data that the interhemispheric gradient is clearly discernible. The Taiwan measurements range between 0.10 ppt and 0.22 ppt, which is consistent with the large variability seen at Taiwan's measurement sites for all other PFCs analysed in this work. Mixing ratios in Taiwan are on average 15 % higher than at Cape Grim. Having discussed the atmospheric trends, the next step is to investigate how global emissions changed over time and what their current status is.





### 3.2 Global Emissions

#### 3.2.1 c-$C_4F_8$ Emissions

The model-derived global c-$C_4F_8$ emissions have changed substantially over time (Fig. 7). An increase in emission rates occurred in the 1980s, peaking around 1.7 Gg yr$^{-1}$ (gigagrams per year) in 1986. A possible contributing source to these emis-

sions might have been the increasing use of liquid PFC coolants in temperature control units within semiconductor processing. The design of these temperature control units, which was initially made to be used with water and glycol, was inadequate to prevent any leakage of the replacement PFCs from the pumps and seals (Tuma and Knoll, 2003; EPA, 2008). The following rapid decrease in emissions between 1985 and 1995 despite the growing demand in the semiconductor industry remains unexplained (Oram et al., 2012), especially because it precedes the Kyoto Protocol and any formal initiative taken by the semiconductor

industry to reduce PFC emissions.

Emissions have subsequently continued to increase from the 1990s minimum. The global emission trend is consistent up to 2006 with our previously reported data by Oram et al. (2012) and agree well on average with the most recent results of Mühle et al. (2019), although our data show larger variability for some periods. However, the extended time series reveals that emission rates do not show any signs of stabilisation and have instead continued to increase, and even accelerate after

$\sim$ 2012. Annual emission are now approaching rates of 2.0 Gg yr$^{-1}$, which are, within uncertainties, comparable to rates determined for the mid-1980s. This compares quite well with the estimated global emissions of 2.2 Gg yr-1 in 2017 by the recent work of Mühle et al. (2019). Emission rates have increased by approximately 50 % since 2010. The increasing emissions are interesting especially in the light of the substantial PFC emission-reduction efforts by the semiconductor industry since 1999 by means of process optimisation, alternative chemistries, PFC recovery, and abatement (Beu, 2005). The report by the

International Sematech Manufacturing Initiative (ISMI) refers to effective and successful reduction efforts for all PFCs by the semiconductor industry, explicitly describing those for $CF_4$, $C_2F_6$, $C_3F_8$, and c-$C_4F_8$ (Beu, 2005). Giving credibility to these reduction efforts by the World Semiconductor Council (WSC), it is perhaps unexpected that c-$C_4F_8$ emissions are increasing at their current rate. However, the list of WSC members is not exclusive to all major semiconductor manufacturers in the world and while technology is becoming more efficient, demand within the electronics industry may be offsetting PFC reduction

efforts. Additionally, c-$C_4F_8$ is one of the candidates as a replacement for lower molecular weight PFCs, such as $CF_4$ and $C_2F_6$, which have longer atmospheric lifetimes and higher GWPs (Tsai et al., 2002). Finally, the results of Mühle et al. (2019) point toward fluoropolymer production as another major source of atmospheric c-$C_4F_8$.

An enormous discrepancy is evident between c-$C_4F_8$ emissions based on reported production (bottom-up) in the EDGARv4.2 data base and emissions derived based on the model simulations of the atmospheric mixing ratios (top-down). This finding con-

tinues a trend noted for our earlier measurements (Oram et al., 2012). The c-$C_4F_8$ emission rates from the EDGARv4.2 database are lower than model-derived values by two orders of magnitude. This reveals the continued lack of reporting by nations on PFC production and emissions. However, a fraction of the discrepancies between atmospherically-derived and report-derived emissions is possibly attributable to inadvertent PFC production and emission, despite efforts to reduce PFC release into the atmosphere through leakage (Beu, 2005). Some of these inadvertent PFC sources may remain unidentified, complicating at-





tempts to locate source regions and source types. Even the recent evidence towards fluoropolymer production sources by Mühle et al. (2019), which is consistent with their much larger observational data set for c-$C_4F_8$, still remains somewhat speculative on a global scale.

### 3.2.2  n-$C_4F_{10}$ and n-$C_5F_{12}$ Emissions

Emission rates for n-$C_4F_{10}$ and n-$C_5F_{12}$ are comparable to each other both in terms of trend and magnitude, which is consistent with previous work (Laube et al., 2012; Ivy et al., 2012b) (Fig. 8). Emissions of both compounds initially rise steadily to peak at around 0.30 Gg yr$^{-1}$ in the mid-1990s, after which they decrease. n-$C_4F_{10}$ and n-$C_5F_{12}$ are used in refrigeration technology and fire extinguishing methods (Robin and Iikubo, 1992; Mazurin et al., 1994; EDGAR, 2014). The upward emission trend prior to the mid-1990s may be attributed to the growth of the electronics industry, which required cooling technology, and the

substitution of ozone-depleting chlorofluorocarbons (CFCs) in refrigeration and fire extinguishing applications. Subsequent replacement strategies of PFC applications after the signing of the Kyoto Protocol in 1997, such as substitution by alternative chemistries, may have triggered the decline in these emissions.

Maximum emission rates reported in Laube et al. (2012) are 0.27 Gg yr$^{-1}$ for n-$C_4F_{10}$ and 0.31 Gg yr$^{-1}$ n-$C_5F_{12}$, which agrees within the uncertainties of the current work. A similar conclusion is valid for the maximum emissions rates in Ivy et al.

(2012) (Fig. 8). In order to fit the observed mixing ratios, emission rates are required to have stabilised in the last decade. In the current work, n-$C_4F_{10}$ stabilises at lower emission rates than in the late 1970s. These trends are also qualitatively comparable to the work by Ivy et al. (2012b). Laube et al. (2012) already suspected a stabilisation in emission rates, which is supported by the extended data set presented in the current study. This low emission rate is consistent with 1) the low NH-SH gradient seen in the Cape Grim and Tacolneston observations, and 2) the Taiwan data only showing moderately elevated mixing ratios.

As with c-$C_4F_8$, it is apparent that top-down and the bottom-up approaches estimate very different annual global emissions for n-$C_4F_{10}$ and n-$C_5F_{12}$ (Fig. 8). It is important to note that data reported in the EDGARv4.2 database do not distinguish between different isomers. Hence, the reported emissions in the EDGARv4.2 database are likely a combination of various isomers. Nevertheless, it is still worthwhile comparing these data on reported emissions to the modelled emissions for the n-isomers in the current study, because the abundance of other isomers besides the n-isomer is very low for $C_4F_{10}$ and not

detectable for $C_5F_{12}$. The discrepancy between the modelled emissions for the isomers in the current study and the total reported emissions in the EDGARv4.2 database is around two orders of magnitude for $C_4F_{10}$ and four orders of magnitude for $C_5F_{12}$, with maximum EDGARv4.2 emission rates of 0.02 Gg yr$^{-1}$ ($C_4F_{10}$) and of $5.26 \times 10^{-5}$ Gg yr$^{-1}$ ($C_5F_{12}$) (EDGARv4.2 data for n-$C_5F_{12}$ is not shown) (Fig. 8).

### 3.2.3  i-$C_6F_{14}$ and n-$C_6F_{14}$ Emissions

One of the most striking features of the top-down derived emission trend for n-$C_6F_{14}$ is the sudden increase in emissions in the mid-1990s (Fig. 9). Emission rates are estimated to be constant at less than 0.2 Gg yr$^{-1}$ up until ∼1994, but then increase by a factor of six to reach 1.21±0.10 Gg yr$^{-1}$ by ∼1997. n-$C_6F_{14}$ is liquid at room temperature and thus has widely used applications as a heat transfer fluid. As described in section 3.2.1, the switch to using PFCs as heat transfer fluids was rapid



due to their effectiveness at regulating heat which was necessary to cope with the increasing demand of the semiconductor manufacturing industry. This development could be an explanation for the rapid onset of n-$C_6F_{14}$ emissions. Additionally, the onset roughly follows the signing of the Montreal Protocol in 1987 and may have triggered the replacement of some CFCs, such as CFC-113, which also had applications in equipment cooling (EPA, 2008). However, the data in this time period are relatively scarce and thus the exact timing of the increase in emissions is not well constrained. By 2013, emission rates have decreased to below 0.2 Gg yr$^{-1}$ again. This reduction in emissions may partly be attributable to the substitution of n-$C_6F_{14}$ as heat-transfer fluid by hydrofluoroethers, which have some superior properties compared to most liquid PFCs (Tuma and Tousignant, 2001).

The effect of separating the $C_6F_{14}$ isomers on the estimated emissions is apparent when comparing them to emission estimates in Laube et al. (2012) where the isomers are not separated (Fig. 9); the emission trend for i-$C_6F_{14}$ is disaggregated from that of the actual n-$C_6F_{14}$ emission trend. Observations presented in Laube et al. (2012) led n-$C_6F_{14}$ emission rate estimates to increase earlier and more gradually and to decline more gradually as well, underestimating the maximum emissions in the late 1990s compared to those in EDGARv4.2 (although they are comparable within the uncertainties) (Fig. 9).

In contrast with n-$C_6F_{14}$, emission rates for i-$C_6F_{14}$ are estimated to have started increasing in 1992 (rather than in 1994, as estimated for n-$C_6F_{14}$, although within the uncertainties of the early data set this is not a significant difference) and its initial rate of increase is not as fast. It reaches a maximum of 0.25±0.02 Gg yr$^{-1}$ in 1996-1997, which is when n-$C_6F_{14}$ emissions reach their maximum values as well. i-$C_6F_{14}$ emissions subsequently decrease gradually and stabilise at about 0.09±0.007 Gg yr$^{-1}$ by 2005. Interestingly, global emissions for i-$C_6F_{14}$ are constant since $\sim$ 2004, but emissions for n-$C_6F_{14}$ seem to continue to decrease until about 2012 (Fig. 5). This is consistent with the increasing i-$C_6F_{14}$ : n-$C_6F_{14}$ ratio in the last two decades (Section 3.1.3).

The current study has shown that the i-isomer of $C_6F_{14}$ is clearly present and increasing in the atmosphere (Section 3.1.3). The emissions reported in Laube et al. (2012) and Ivy et al. (2012a) (which are based on dilutions of the n-isomer calibrated against a working tank containing compressed, unpolluted air that inevitably contains both isomers) cannot be directly compared to the emissions derived here for the isomers separately. Similarly, due to the fact that EDGARv4.2 does not distinguish PFC isomers, caution must be taken when comparing the emission estimates for the $C_6F_{14}$ isomers from the model to the emission reported for the $C_6F_{14}$ compound in EDGARv4.2.

Thus, it would be more accurate to compare the sum of the observation-derived emissions of the separate isomers of $C_6F_{14}$ to the reported EDGARv4.2 data. This results in three major improvements. One, it reduces the gap between derived emissions and EDGARv4.2 emissions prior to 1995, as seen for n-$C_6F_{14}$ emission estimates. Two, peak emission rates have increased from 1.21±0.1 Gg yr$^{-1}$ to 1.46±0.12 Gg yr$^{-1}$ and thereby exceed EDGARv4.2 values. However, given that reported emission rates in EDGARv4.2 inherently also have an uncertainty that is not shown here and that it is more likely that nations underreport emissions, this increase in peak emissions is probably realistic. Three, the sum of the estimated emission rates of both $C_6F_{14}$ isomers agrees exceedingly well with the EDGARv4.2-reported values from 1999 onwards.


### 3.2.4 n-$C_7F_{16}$ Emissions

Emission rate estimates for n-$C_7F_{16}$ increase from $0.05\pm0.003$ Gg yr$^{-1}$ in 1980 to a maximum of $0.18\pm0.01$ Gg yr$^{-1}$ in 1985. Emissions remain stable at this emission level until 2017 (Fig. 10), despite emission-reduction efforts for higher molecular weight PFCs as heat transfer fluids by leak proofing pumps and applying alternative chemistry (EPA, 2008). This suggests that
n-$C_7F_{16}$ either has properties and/or applications that are challenging to substitute and/or unknown atmospheric sources.

    Similar to Laube et al. (2012), model-based emission estimates are higher compared to the EDGARv4.2 reported values before 1990 and after 1999, and are lower in between. Maximum emission rates for n-$C_7F_{16}$ derived by Laube et al. (2012) are $0.23\pm0.1$ Gg yr$^{-1}$. This slightly higher maximum as compared to the results in this work are likely due to calibration differences as a) at least two minor isomers of $C_7F_{16}$ are present in the atmosphere and were not chromatographically separated
by Laube et al. and b) the n-$C_7F_{16}$ calibration in Laube et al. was not based on a pure isomer, but a technical mixture of isomers. The uncertainties are larger in the work by Laube et al., due to the additional uncertainty that had to be attributed based on the limited purity of $C_7F_{16}$ used for the calibration.

    The estimated emissions have a distinctly different trend compared to those in Ivy et al. (2012b), which seem to be in better agreement with the peak seen in the 1990s in the EDGARv4.2 data. Their larger data set, which also included time series in
both the Northern and Southern Hemisphere, is able to better constrain their model. This results in more detailed features in the simulated atmospheric trends. However, to assume that the measured $C_7F_{16}$ in previous work only consists of n-$C_7F_{16}$ would overestimate n-$C_7F_{16}$ mixing ratios by an unknown amount (our calibration changed by 11 %, see Section 2.3), because even though previous work diluted high purity compounds in calibration work, the amount of likely unseparated isomers present in their working standard tanks and samples remains unknown.

Observations of the continuing increasing background mixing ratios in the Southern Hemisphere strongly suggest that global emissions of n-$C_7F_{16}$ persist at an approximately constant rate. These conclusions remains consistent with those made in Laube et al. (2012) and Ivy et al. (2012b) for $C_7F_{16}$.

### 3.3 Possible Source Regions and Source Types

The high variability of the PFC mixing ratios in Taiwan described in this work suggests that these measurement sites are close
to major PFC sources. The fact that these sampled air masses are not yet well mixed offers the possibility to investigate the following questions: 1) Which PFCs are likely co-emitted and have similar sources? 2) What are likely source regions for these PFCs within Asia? 3) And what are likely source types of the PFCs measured in Taiwan? To pursue the answers to these, the air measurements are combined with the NAME model data, as described in section 2.4.

    The similarity of sources among the PFCs can be studied by looking at the inter-species correlations (Fig. S3). These results
show that all PFCs, including $C_2F_6$ and $C_3F_8$ ($CF_4$ was not measured), are significantly correlated with each other, with squared Spearman coefficients ranging from 0.14 to 0.68 (Table S3). The best correlations are observed between $C_2F_6$ and $C_3F_8$ (in contrast to what was reported by Zhang et al. (2017)) and between c-$C_4F_8$ and n-$C_4F_{10}$ (Table 3). i-$C_6F_{14}$ and n-$C_6F_{14}$ are not very well correlated, which supports the hypothesis of at least partly independent sources.





Generally, the PFCs correlate less with with any of the 15 regions used to quantify sources of particle densities used in the dispersion modelling, although sample numbers limit the statistics (Table S4). The best correlations are observed for East China for all compounds with the highest squared Spearman rank coefficients found for n-$C_7F_{16}$ (0.49) and n-$C_5F_{12}$ (0.47) (Table 3). Significant correlations are also found for all PFC species with the East China Sea, which can be explained by the

fact that air masses from East China travel over the East China Sea before they reach Taiwan.

All PFCs correlate well with energy industry and domestic sources, which is linked to population density (Table S5). Other best coefficients are found either with the signature of power plants ($C_2F_6$, $C_3F_8$, c-$C_4F_8$, i-$C_6F_{14}$, n-$C_6F_{14}$) or solvents (n-$C_4F_{10}$, n-$C_5F_{12}$, n-$C_7F_{16}$) (Table 3). Some correlation plots seem to show bifurcation within the scattered data (most notably in the correlations of most PFCs with $C_3F_8$), suggesting that multiple source signatures may exist (Fig. S3).

The combination of the NAME modelling work and air measurements shows potential to identify and quantify regional PFC sources and source types. However, the interpretation of the results may be challenging, such as the possibility of the heterogeneity in emissions over time (e.g. discontinuous emissions may lead to two similar particle density signatures linked to two very different atmospheric concentrations measured in Taiwan). The collection of air samples at the sites in Taiwan is ongoing. New data in addition to the current dataset can be used to inspect PFC source regions and types more rigorously,

including the significance of the initial signs of any bifurcation in the correlations between PFCs and CO concentrations that share source types. Nevertheless, this regional study on Taiwan is indicative of the relation between measured atmospheric PFC concentrations and important economic sectors, which feeds back to the implications on rising global PFC trends.

## 4 Implications

The changing trends and emissions of six PFCs have been thoroughly discussed in Sections 3.1 and 3.2, but since these

greenhouse gases only occur at very low ppt and sub-ppt levels in our atmosphere, the significance of their influence on our climate system needs further clarification. In order to gain more insight into this matter, the annual global emission rates of all six PFCs covered in this work have been converted to the equivalent of $CO_2$ by using the updated GWPs from the latest IPCC report (Myhre et al., 2013) (Table 1). As isomers have the same molecular mass, the GWP for i-$C_6F_{14}$ is estimated based on the GWP reported for n-$C_6F_{14}$ (7910; (Myhre et al., 2013) and scaled with the ratio of the radiative efficiencies of i-$C_6F_{14}$

(0.41 $Wm^{-2}$ $ppbv^{-1}$; Bravo et al. (2010)) and n-$C_6F_{14}$ (0.44 $Wm^{-2}ppbv^{-1}$; (Myhre et al., 2013). PFCs have extremely long atmospheric lifetimes, and thus their $CO_2$ equivalents are calculated here as accumulating over time (Fig. 11) (Laube et al., 2012). By 2017, the cumulative global emissions of c-$C_4F_8$, n-$C_4F_{10}$, n-$C_5F_{12}$, i-$C_6F_{14}$, n-$C_6F_{14}$, and n-$C_7F_{16}$ amounted to 833 million metric tonnes of $CO_2$ equivalents. This represents an increase of 23 % between the beginning of 2010 and the end of 2017. 61 % of the total $CO_2$ equivalent is attributable to c-$C_4F_8$. The difference between the total $CO_2$ equivalent by the end

of 2009 reported in Laube et al. (2012) (~750 million metric tonnes $CO_2$ equivalent) and in the current work (~678 million metric tonnes $CO_2$ equivalent) is partly the result of the separation of isomers, but mostly due to the updated global warming potentials (Myhre et al., 2013).





The importance of PFCs extends beyond the scope of climate in the troposphere. The inert nature and continuously-increasing concentrations in the troposphere of PFCs over time make them interesting as potential new age-of-air (AoA) tracers. AoA tracers are crucial for research on stratospheric circulation (e.g., Engel et al., 2017) and troposphere-stratosphere fluxes (e.g., Bönisch et al., 2009), which enables further understanding of threats to the ozone layer. $SF_6$ and $CO_2$ are among the most commonly used AoA tracers, but have limitations that significantly compromise conclusions on dynamic and chemical stratospheric processes (Stiller et al., 2012; Ray et al., 2017; Leedham Elvidge et al., 2018). Some of the lower molecular weight PFCs, such as $CF_4$, $C_2F_6$, $C_3F_8$, have already been shown to be suitable age tracers (Leedham Elvidge et al., 2018). Considering all PFCs presented in the current work, c-$C_4F_8$ has a particular potential to be an excellent age tracer for three reasons: 1) It has a well-constrained time series from 1978 to the present. 2) It still has a sufficiently fast tropospheric growth rate (3.2 ± 0.2% on average between 2010 and 2018). 3) Observations for background c-$C_4F_8$ levels in the Southern Hemisphere have an average measurement uncertainty of about 1 % (1.0±0.7 %). The other PFCs presented in this work are currently not suitable as AoA tracers as their annual growth rates since 2010 (∼0.8-2.9%) do not exceed their average measurement precisions (∼3.1-7.3%).

## 5 Conclusions

This work has extended existing time series on observations at Cape Grim, Australia, for atmospheric mixing ratios of c-$C_4F_8$, n-$C_4F_{10}$, n-$C_5F_{12}$, n-$C_6F_{14}$, and n-$C_7F_{16}$ from 2010 to 2018 and converted it onto an improved calibration scale. The calibration scale is improved due to the separation of isomers, which leads to a more accurate analysis of tropospheric trends. Moreover, the background trend in i-$C_6F_{14}$ mixing ratios in the Southern Hemisphere is reported here for the first time and some of the other PFCs have been measured of the first time as discrete isomers.

The background mixing ratios for all six PFCs continue to increase and reached 1.51 ppt (c-$C_4F_8$), 0.18 ppt (n-$C_4F_{10}$), 0.15 ppt (n-$C_5F_{12}$), 0.07 ppt (i-$C_6F_{14}$), 0.22 ppt (n-$C_6F_{14}$), and 0.11 ppt (n-$C_7F_{16}$) by the end of 2017. An increasing trend is observed most clearly for atmospheric background concentrations of c-$C_4F_8$, i-$C_6F_{14}$, and n-$C_7F_{16}$, which have increased by 27 %, 19 %, and 21 % in the last eight years, respectively. Atmospheric mixing ratios for n-$C_4F_{10}$, n-$C_5F_{12}$, and n-$C_6F_{14}$ have increased by 9 %, 6 %, and 9 %, respectively.

Southern Hemispheric background trends are compared to mixing ratios in the Northern Hemisphere. Air samples from the Tacolneston site in the UK show an interhemispheric gradient for c-$C_4F_8$ and n-$C_7F_{16}$, indicating substantial emissions within the Northern Hemisphere. This is consistent with their faster (as compared to the other four PFCs) increasing atmospheric levels observed in unpolluted, Southern Hemispheric air.

All PFC mixing ratios in Taiwan air samples are extremely variable and show frequent enhancements. These observations show that the Taiwan measurement sites are in close proximity to substantial PFC sources, which is not surprising given the extensive presence of electronic manufacturing industry in East Asia, which is believed to be one of the main atmospheric sources of these gases (Beu, 2005; Saito et al., 2010).



Finally, emission rate model-estimates generally agree well with previous modelling work. Most noteworthy is the continuation of the increasing trend for c-$C_4F_8$ emissions. This increase reflects an acceleration of mixing ratio increases in recent years. Our work provides an independent verification of the recent trend of that particular gas and is largely in agreement with the findings of the more extensive c-$C_4F_8$-focused work of Mühle et al. (2019). For the longer-chain PFCs, differences with

results by Laube et al. (2012) are mainly due to the improved calibration scales. Especially the comparison for $C_6F_{14}$ has improved as its two main isomers are now separated.

The current study demonstrates that for most of the six PFCs in this work, the emissions determined with the bottom-up approach of the EDGARv4.2 database are much lower than those determined by the top-down approach using observations of mixing ratios. This suggests that production and emissions of PFCs are going unreported, which demonstrates the need to

better determine the locations of major PFC sources. Analysis of Taiwan PFC mixing ratios and simulated particle dispersion and CO concentrations by the NAME model indicate the potential of such regional studies in understanding the global trends. This analysis will gain statistical weight with more measurements added to the dataset each year and is therefore considered ongoing. Voluntary PFC reduction plans exists among semiconductor industry associations (Beu, 2005). Even though these may be adhered to, PFC emissions are not reported by most nations, including those in East Asia (Saito et al., 2010). Determining

the contribution of various regions to rising PFC mixing ratios thus remains a challenge.

Monitoring and regulating PFC emissions is relevant to future climate, as illustrated by the cumulative $CO_2$ equivalents of over 830 million tonnes for all PFCs reported here (an increase of 155 million tonnes between the beginning of 2010 and the end of 2017), and the potential to be applied in other areas of atmospheric research, such as circulation and chemistry changes in the stratosphere.



*Author contributions.* ESD led the manuscript writing process and carried out the emission modelling and some of the measurements. KEA, LJG, AJH, ELE and JCL also contributed to the measurements while CER, MJA, ZF, NMH, and MP contributed to the modelling parts of the study. CC, PJF, RLF, SO, DEO, CO and WTS contributed through the coordination and execution of the various sampling activities. JCL developed the concept for this study and all authors contributed to developing it further as well as to the manuscript.

*Acknowledgements.* This work was supported by the European Research Council's (ERC) funding for the EXC³ITE project (EXploring stratospheric Chemistry, Composition, and Circulation using Innovative TEchniques). The collection and curation of the Cape Grim Air Archive is jointly funded by CSIRO, the Bureau of Meteorology (BoM) and Refrigerant Reclaim Australia; BoM/CGBAPS staff at Cape Grim were/are largely responsible for the collection of archive samples and UEA flask air samples; the original (mid-1990s) subsampling of the archive for UEA was funded by AFEAS and CSIRO, with ongoing subsampling by CSIRO. Operation of Tacolneston is funded by the
Department of Business, Energy & Industrial Strategy (BEIS) through contract TN 1537/06/2018. We would specifically like to thank Stephen Humphrey and Andy MacDonald for all their work collecting the samples at the Tacolneston site and transporting them to UEA. Taiwan-related work was supported by the NERC IOF award NE/J016012/1. This work used the NAME atmospheric dispersion model, for which the UK Met Office provided the NWP meteorological datasets. Karina E. Adcock's PhD is supported by the Natural Environment Research Council through the EnvEast Doctoral Training Partnership [PhD Studentship NE/L002582//1]. Norfazrin Mohd Hanif's PhD is supported by
the Ministry of Education Malaysia (MOE) and Universiti Kebangsaan Malaysia (UKM). Johannes C. Laube and Lauren J. Gooch received funding from the UK Natural Environment Research Council (Research Fellowship NE/I021918/1 and studentship NE/1210143) and David E. Oram from the National Centre for Atmospheric Science.





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

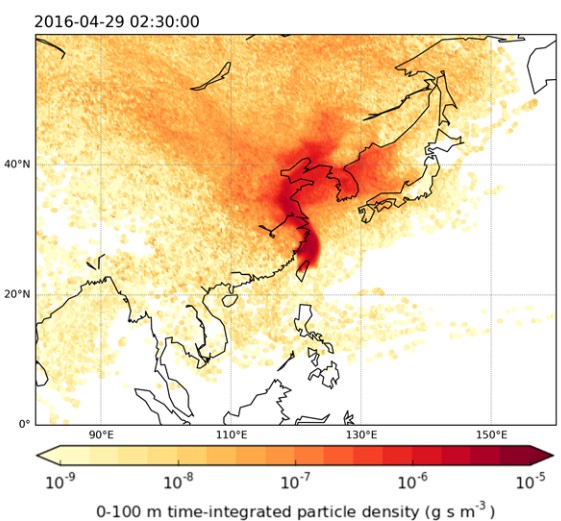

**Figure 1.** Example of the particle footprints, indicating where air sampled in Taiwan came from. The colour scale is logarithmic and represents the calculated time-integrated particle density (gs m$^{-3}$) within the surface layer (0-100 m) during 12 days prior to the sampling days given a point release at Taiwan of gs$^{-1}$. Darker colours indicate a greater influence of the region to the chemical composition of the air sampled in Taiwan compared to the lighter colours.





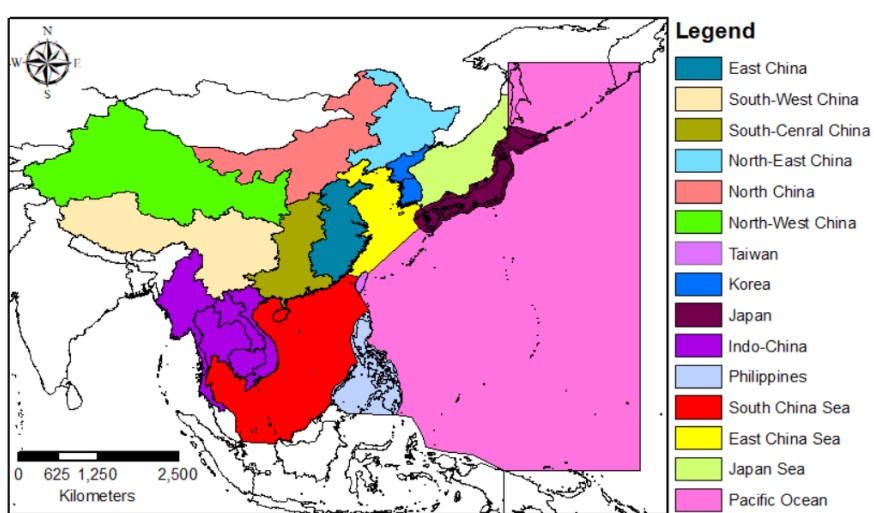

**Figure 2.** Regions for which the contribution to the footprint simulated by the NAME model is quantified.

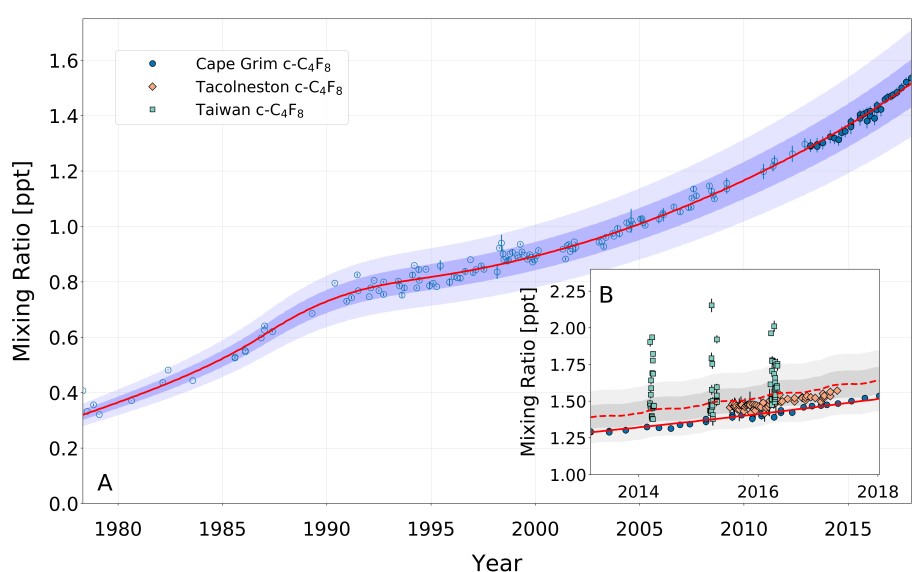

**Figure 3.** A) Mixing ratios at Cape Grim of c-$C_4F_8$ between 1978 and 2018. Data prior to 2010 are shown as empty symbols (Oram et al., 2012), while data from samples collected after 2010 are shown as filled symbols to illustrate the part of the time series that is extended in the current work. The atmospheric trend simulated by the model is represented by the red line. Total uncertainties (light blue) and trend uncertainties (dark blue) are indicated with the shaded areas along the trend line. B) Mixing ratios after 2010 for Cape Grim (circles), for Tacolneston (diamonds), and Taiwan (squares). The red line indicates the modelled Southern Hemisphere baseline trend at Cape Grim, while the dashed red line indicates the modelled trend for mixing ratios at Tacolneston. Total and trend uncertainties are indicated with the shaded grey areas along the Tacolneston dashed-trend line.

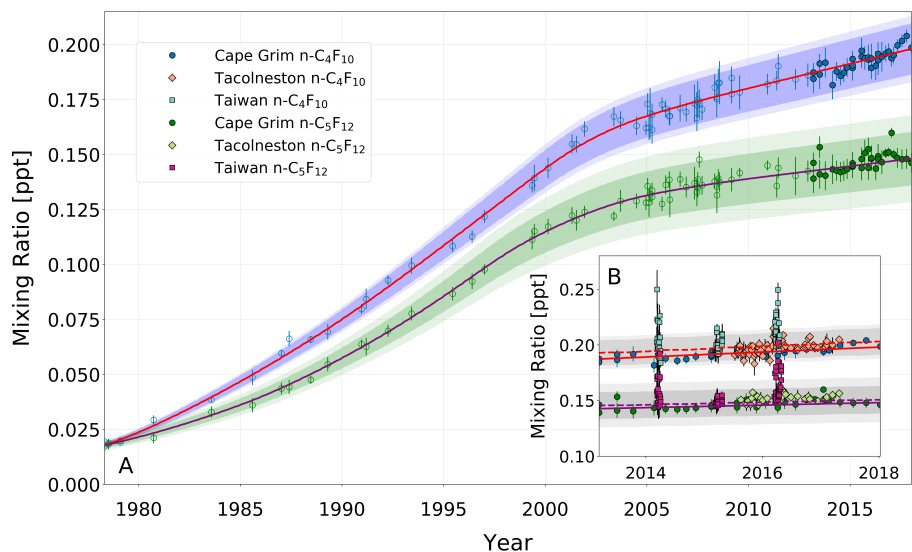

**Figure 4.** A) Mixing ratios at Cape Grim of n-C$_4$F$_{10}$ (blue circles) and n-C$_5$F$_{12}$ (green circles) between 1978 and 2018. Data prior to 2010 are shown as empty symbols (Laube et al., 2012), while data from samples collected after 2010 are shown as filled symbols to illustrate the part of the time series that is extended in the current work. Note that the data from Laube et al. have been converted to a new and improved calibration scale. The atmospheric trends simulated by the model are represented by the red line for n-C$_4$F$_{10}$ and by magenta line for n-C$_5$F$_{12}$. Total uncertainties (light blue for n-C$_4$F$_{10}$, light green for n-C$_5$F$_{12}$) and trend uncertainties (dark blue for n-C$_4$F$_{10}$, dark green for n-C$_5$F$_{12}$) are indicated with the shaded areas along the trend line. B) n-C$_4$F$_{10}$ mixing ratios after 2010 for Cape Grim (blue circles), for Tacolneston (orange diamonds), and Taiwan (light blue squares); n-C$_5$F$_{12}$ mixing ratios after 2010 for Cape Grim (green circles), for Tacolneston (light green diamonds), and Taiwan (magenta squares). The red and magenta lines indicate the modelled Southern Hemisphere baseline trend at Cape Grim for n-C$_4$F$_{10}$ and n-C$_5$F$_{12}$, respectively; while the dashed red and magenta lines indicate the modelled trend for mixing ratios at Tacolneston for n-C$_4$F$_{10}$ and n-C$_5$F$_{12}$, respectively. Total and trend uncertainties are indicated with the shaded grey areas along the Tacolneston dashed-trend lines for both n-C$_4$F$_{10}$ and n-C$_5$F$_{12}$.

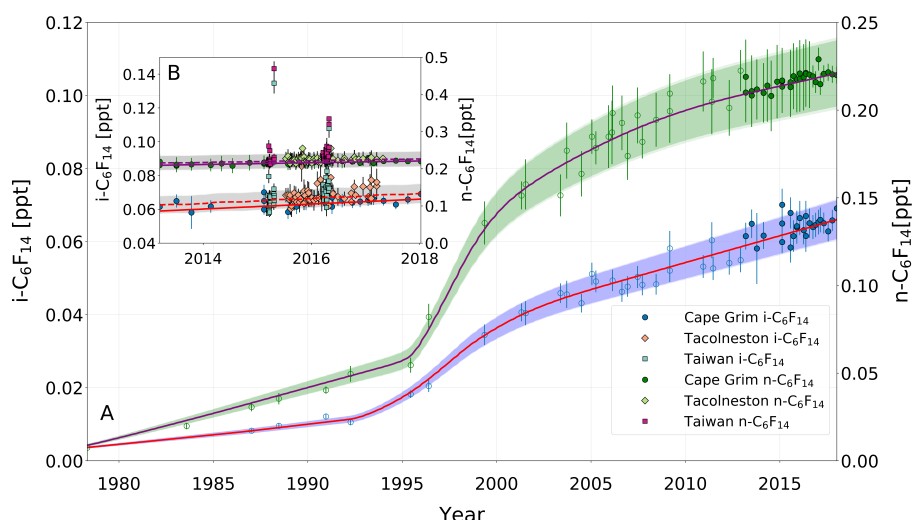

**Figure 5.** A) Mixing ratios at Cape Grim of i-$C_6F_{14}$ (blue circles) and n-$C_6F_{14}$ (green circles) between 1978 and 2018. Note that data for n-$C_6F_{14}$ are plotted on a secondary axis. Data prior to 2010 are shown as empty symbols (Laube et al., 2012), while data from samples collected after 2010 are shown as filled symbols to illustrate the part of the time series that is extended in the current work. Note that the data from Laube et al. have been converted to a new and improved calibration scale. The atmospheric trends simulated by the model are represented by the red line for i-$C_6F_{14}$ and by magenta line for n-$C_6F_{14}$. Total uncertainties (light blue for i-$C_6F_{14}$, light green for n-$C_6F_{14}$) and trend uncertainties (dark blue for i-$C_6F_{14}$, dark green for n-$C_6F_{14}$) are indicated with the shaded areas along the trend line. B) i-$C_6F_{14}$ mixing ratios after 2010 for Cape Grim (blue circles), for Tacolneston (orange diamonds), and Taiwan (light blue squares); n-$C_6F_{14}$ mixing ratios after 2010 for Cape Grim (green circles), for Tacolneston (light green diamonds), and Taiwan (magenta squares). The red and magenta lines indicate the modelled Southern Hemisphere baseline trend at Cape Grim for i-$C_6F_{14}$ and n-$C_6F_{14}$, respectively; while the dashed red and magenta lines indicate the modelled trend for mixing ratios at Tacolneston for i-$C_6F_{14}$ and n-$C_6F_{14}$, respectively. Total and trend uncertainties are indicated with the shaded grey areas along the Tacolneston dashed-trend lines for both i-$C_6F_{14}$ and n-$C_6F_{14}$.

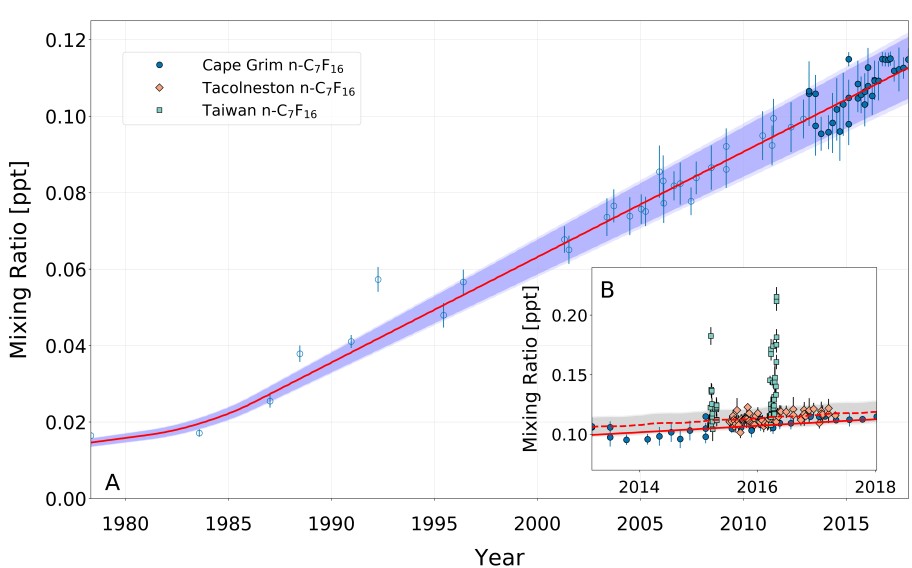

**Figure 6.** Same as Fig. 3, but for n-$C_7F_{16}$. Data from Laube et al. (2012) are plotted as empty markers. Note that the data from Laube et al. have been converted to a new and improved calibration scale.

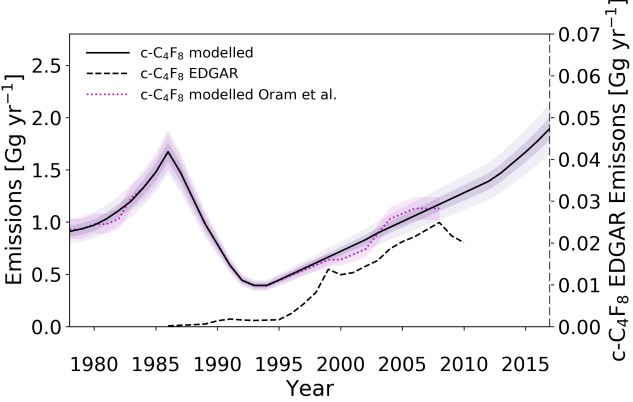

**Figure 7.** Estimated global emission rates for c-$C_4F_8$ in this study (full black line), in Oram et al. (2012) (dotted magenta line), and as reported in the EDGARv4.2 database (dashed black line). Note that the EDGARv4.2 emission rates are plotted on the secondary axis. Uncertainty envelopes include contributions from measurements, modelling, and calibrations.

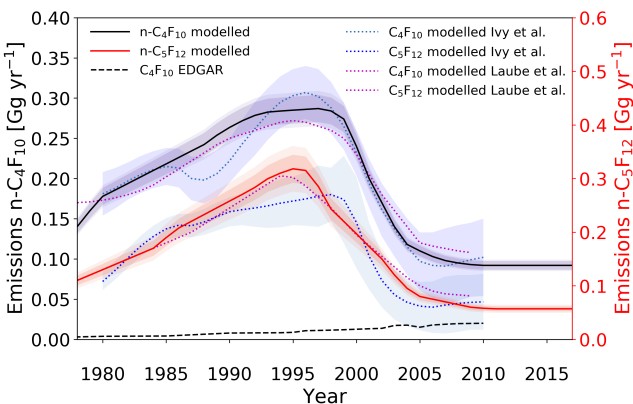

**Figure 8.** Estimated global emission rates for n-C$_4$F$_{10}$ and n-C$_5$F$_{12}$ (full red and black lines) in this study, global emission rates for C$_4$F$_{10}$ as reported in the EDGARv4.2 database (dashed black line; no distinction between isomers), modelled emissions by Ivy et al. (2012b) for C$_4$F$_{10}$ and C$_5$F$_{12}$ (dotted light and dark blue-hued lines, respectively; no distinction between isomers), and modelled emissions by Laube et al. (2012) for C$_4$F$_{10}$ and C$_5$F$_{12}$ (dotted red and magenta lines, respectively; no distinction between isomers). Note that the maximum EDGARv4.2 emission rate for n-C$_5$F$_{12}$ is less than $0.6 \times 10^{-4}$ and is therefore not plotted here. Shadings illustrate the uncertainties.


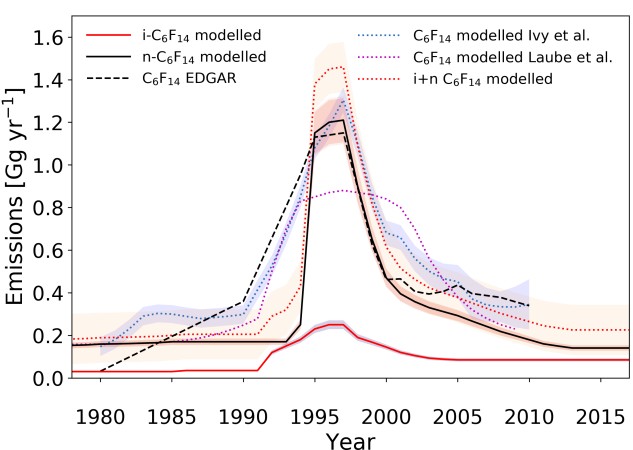

**Figure 9.** Estimated global emission rates for i-$C_6F_{14}$ and n-$C_6F_{14}$ in this study (full red and black lines, respectively); and global emission rates for $C_6F_{14}$ as reported in the EDGARv4.2 database (dashed black line; no distinction between isomers), by Ivy et al. (2012b) (dotted blue line; no distinction between isomers), and by Laube et al. (2012) (dotted magenta line; no distinction between isomers). The sum of the global emissions for the isomers of $C_6F_{14}$, as reported in the current work, is illustrated by the dotted orange line. Shadings illustrate uncertainties.



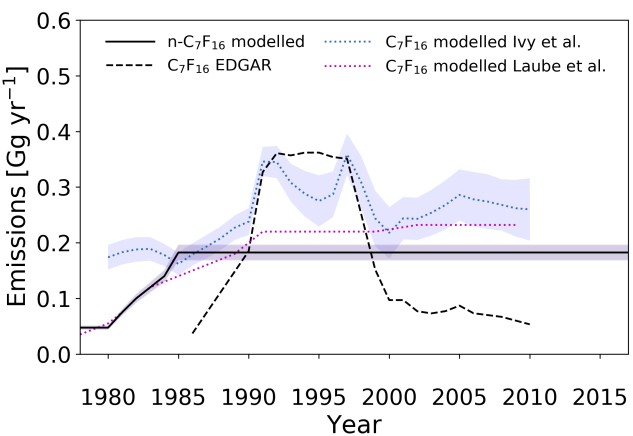

**Figure 10.** Estimated global emission rates for n-$C_7F_{16}$ in this study (full black line) and global emission rates for $C_7F_{16}$ as reported in the EDGARv4.2 database (dashed black line; no distinction between isomers), as reported in Ivy et al. (2012b) (dotted blue line; no distinction between isomers), and as reported in Laube et al. (2012) (dotted magenta line; no distinction between isomers).





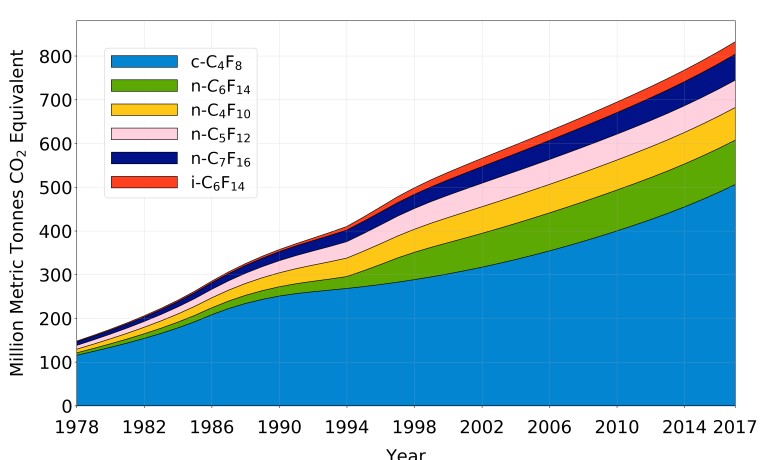

**Figure 11.** $CO_2$ equivalent [million metric tonnes] of c-$C_4F_8$, n-$C_4F_{10}$, n-$C_5F_{12}$, i-$C_6F_{14}$, n-$C_6F_{14}$, n-$C_7F_{16}$ based on emissions between 1978 and 2017.



**Table 1.** Atmospheric lifetimes and global warming potentials (GWP) on a 100 year time horizon as reported in the 2013 Intergovernmental Panel on Climate Change (IPCC) report for all PFCs discussed here (Myhre et al., 2013).

| PFC | Lifetime [yr] | GWP, 100 yr time horizon |
|---|---|---|
| c-$C_4F_8$ | 3200 | 9540 |
| n-$C_4F_{10}$ | 2600 | 9200 |
| n-$C_5F_{12}$ | 4100 | 8550 |
| i-$C_6F_{14}$ | 3100[a] | 7370[b] |
| n-$C_6F_{14}$ | 3100 | 7910 |
| n-$C_7F_{16}$ | 3000 | 7820 |

[a] Lifetime for i-$C_6F_{14}$ is assumed to be the same as for n-$C_6F_{14}$. [b] GWP for i-$C_6F_{14}$ is estimated based on the radiative efficiency reported in Bravo et al. (2010).





**Table 2.** Details on locations of sampling sites and dates, gas-chromatography columns used, and number of samples of which the measurements for each PFC compound are reported here. Differences among number of samples per compound are due to a combination of the length of the period during which the compound was measured, the chromatography method, and the precision of the measurement affected by baseline distortions.

| Site | Location | Dates | CG Column | Compounds and Samples | | | | | |
|---|---|---|---|---|---|---|---|---|---|
| | | | | $c\text{-}C_4F_8$ | $n\text{-}C_4F_{10}$ | $n\text{-}C_5F_{12}$ | $i\text{-}C_6F_{14}$ | $n\text{-}C_6F_{14}$ | $n\text{-}C_7F_{16}$ |
| Cape Grim | Tasmania, Australia, 41 °S, 145 °E | 1978- January 2018 | CP Plot | 150 | 90 | 89 | 53 | 62 | 62 |
| Tacolneston | Norfolk, UK, 52 °N, 1 °E, 185 m tower | July 2015 - April 2017 | CP Plot | 60 | 60 | 34 | 54 | 60 | 59 |
| Taiwan | Cape Fuguei, 25 °N, 122 °E | March and April 2014, 2016 | GasPro (2014), CP Plot (2016) | 93 | 79 | 86 | 61 | 61 | 61 |
| | Hengchun, 22 °N, 121 °E | March and April 2013, 2015 | GasPro (2013), CP Plot (2015) | | | | | | |





**Table 3.** An overview of the best Spearman rank correlations values (squared) for all PFCs measured in Taiwan and reported here, including $C_2F_6$ and $C_3F_8$, with PFC species, regions, and sources. EC: East China, ECS: East China Sea, SCC: South-central China. A complete overview of correlation coefficients can be found in the supplement.

| Species | Best correlation (Spearman rank-squared) with | | | | | |
|---|---|---|---|---|---|---|
| | Species | | Region | | Source | |
| $c$-$C_4F_8$ | $n$-$C_4F_{10}$ (0.62) | $n$-$C_5F_{12}$ (0.56) | EC (0.33) | ECS (0.19) | Domestic (0.42) | Energy (0.41) |
| $n$-$C_4F_{10}$ | $c$-$C_4F_8$ (0.62) | $n$-$C_5F_{12}$ (0.56) | EC (0.20) | ECS (0.12) | Solvents (0.21) | Domestic (0.20) |
| $n$-$C_5F_{12}$ | $c$-$C_4F_8$ (0.56) | $n$-$C_7F_{16}$ (0.49) | EC (0.47) | ECS (0.33) | Solvents (0.50) | Domestic (0.45) |
| $i$-$C_6F_{14}$ | $C_2F_6$ (0.52) | $n$-$C_6F_{14}$ (0.40) | EC (0.33) | ECS (0.19) | Domestic (0.33) | Energy (0.31) |
| $n$-$C_6F_{14}$ | $C_2F_6$ (0.52) | $C_3F_8$ (0.50) | ECS (0.27) | EC (0.18) | Energy (0.26) | Domestic (0.22) |
| $n$-$C_7F_{16}$ | $c$-$C_4F_8$ (0.55) | $C_2F_6$ (0.54) | EC (0.49) | ECS (0.21) | Domestic (0.46) | Solvents (0.44) |
| $C_2F_6$ | $C_3F_8$ (0.68) | $n$-$C_7F_{16}$ (0.54) | EC (0.19) | SCC (0.13) | Energy (0.34) | Domestic (0.25) |
| $C_3F_8$ | $C_2F_6$ (0.68) | $n$-$C_6F_{14}$ (0.54) | EC (0.10) | ECS (0.07) | Energy (0.31) | Transport (0.19) |





**Table 4.** Mixing ratios determined in the working standard, precisions of the calibrations, and accuracies of CFC-11 calibrations compared to NOAA scales. For details on the calibration of c-$C_4F_8$ and n-$C_5F_{12}$, consult Oram et al. (2012) and Laube et al. (2012).

| PFC | Mixing Ratio [ppt] | Analytical Precision [%] | Accuracy for CFC-11 [%] | Calibration Scale Uncertainty [%] |
|---|---|---|---|---|
| c-$C_4F_8$ | $1.565^b$ | - | $<4.60^c$ | $7^c$ |
| n-$C_4F_{10}$ | $0.201^a$ | 0.61-5.04 | -4.19 - -5.47 | 1.75 |
| n-$C_5F_{12}$ | $0.152^b$ | - | $-1.4 - 2.8^d$ | $5.13^d$ |
| i-$C_6F_{14}$ | $0.067^a$ | 0.22-1.31 | -3.96 - -4.58 | 0.52 |
| n-$C_6F_{14}$ | $0.224^a$ | 1.06-2.41 | -1.78 - -5.47 | 0.74 |
| n-$C_7F_{16}$ | $0.115^a$ | 1.52-2.65 | -3.60 - -4.58 | 1.02 |

[a] Mixing ratios determined in working standard by calibrating pure compounds. [b] Mixing ratios determined in working standard by converting calibrated mixing ratios in the previous working standard to current working standard. [c] (Oram et al., 2012). [d] (Laube et al., 2012).