# Peer review of "Trends and Emissions of Six Perfluorocarbons in the Northern and Southern Hemisphere"

_Atmospheric Chemistry and Physics, 2019_

## Referee Comment (RC1) · Anonymous Referee #1 · 2 Dec 2019

In this study, Droste et al present improved analytical capabilities to separate isomer of long-lived perfluorocarbons (PFCs). A few years of updated measurements of PFCs from three sample sets are given and complemented with older published work. Global emissions are derived and compared to inventory-based and earlier observation-based emissions estimates. The work presents some im-portant advancements in analytical techniques and important long-term observations and emissions estimates. Source regions and source types are also discussed here. The manuscript is generally well written and understandable.

My major comments are on some clarifications related to primary calibration, more quantitative comparison to recent work by Laube et al, the publishing on the actual measurement results, and Figure improvments. These are interspersed into the gen-

eral comments below.

Abstract, p. 1, l. 5: Perhaps add 'analytically', to clarify what 'separation' is discussed here (analytically separated from their isomers . . .)

p. 2, l. 4: is 'metric' necessary (isn't ACP using SI units anyway?).

p. 2, l. 6 'despite what?', what does 'this' reference to?

p. 2, l. 5: Does the two-thirds refer to the emissions of the last 8 years or to the emissions since 1978?

p. 3, l. 18: is decofluorobutane. Spelling here different to abstract

p. 3, l. 19: hexodecafluoroheptane. Spelling here different to abstract.

p. 3, l. 21: same issue for perfluoro-2-methylpentane. Suggest to recheck spelling of all chemicals and use consistently throughout.

p. 3, l. 24. '. . . of these. . .' would strictly refer to c-C4F8 and n-C5F12, but I presume that the author means all compounds handled in this manuscript.

p. 3, l. 24. In Table 1, there is a compound with a GWP outside the range indicated here (i-C6F14).

p. 4 objective 2: There is something fundamentally unclear here. Why were new calibration scales needed? I understand that with new methods, isomers could now be separated. However, were the old calibration scales not made with pure reagents, e.g pure n-C4F10, no i-C4F10? If so, then why weren't the old calibration reference standards used for the present work, and the air sample data corrected for the fact that now the two isomers are separated? It appears that the old calibration reference standards would still be good, but the measurement of the air samples would need to be redone or the old ones corrected. Was the same raw material used to produce the new scales. How many cali-bration reference standards were produced — just one?

p. 5, l. 14 – 16. It seems that this was already explained in the Intro objective 1.

p. 6, l. 7: Were compounds used in this study also affected? If not, perhaps be more specific by saying ..'number of compounds not used in this study'.

p. 7, second para. Comment here is similar to p. 4, objective 2.: '..from the observe isomer' observed where – in the old primary calibration standards? If there was i-C4F10 in the old standards, why wasn't it precisely quantified by measuring it vs the new cal standards, then allowing to separate the <2.8% difference into the two factors 'leak-tightness' and 'i-C4F10' influence. Line 9. Lack of observed iso-mers where?

P. 7 l. 12: 'The confidence in the complete separation of all isomers for C6F14 and C7F16 . . .' This sen-tence is in conflict with earlier mentioning of inabilities of separat-ing all isomers of e.g. C7F16.

p. 7, l. 19: '. . . did not differ significantly. . .' seems incorrect, According to Fig. S2 they differ very strongly and significantly and are rather concerning.

p. 7, l. 27. The abbreviation does not seem to match the first letters of the full name.

p. 7, l. 30: Could you provide a guess why there is a difference between the boot-strapped CFC-11 and that measured on the NOAA scale. There is a lot of plural men-tioning, but isn't it only one compari-son, that of the (single) working tank vs the primary calibration tanks?

p. 7, l. 31. How can an accuracy be negative (-5.5%).

p. 7, l. 31: It is incorrect to say that the improvement by the new calibration scale is significant. The improvement is max 2.8% as stated earlier. What improved a lot are the measurement results of the field samples, because i-C4F10 is now excluded from the n-C4F10 measurements. Again, this is mainly a correction of older air measurements, not an improvement of a scale.

p. 7, l. 34: How did you get to the 20% and 11%. Perhaps explain in more detail in

the supplement. Labeling a calibration scale 'new' calls for problems down the road. It would be much clearer to give it a name.

p. 8, l. 4: 'horizontal'. Shouldn't this be 'vertical'?

p. 8, l. 7: Why would a 'not fully represented' atmospheric layer justify leaving it out, there is a good part of the stratosphere in the model.

p. 10, l. 10: Provide some more intercomparison for the c-C4F8 mixing ratio of this study and that of Muhle et al., 2019, either here or in the supplement. Add the Muhle et al. CGAA mixing ratio to the present results. Are the deviations a constant ratio or are there potential nonlinearity issues. Can you derive a calibration scale factor between the UEA and the SIO scales, perhaps based on the CGAA re-sults, or other ways of intercomparisons (Tacolneston?). A well-derived conversion factor is extremely useful e.g. for users of the many data sets presented here and in Muhle et al.

p. 12, l. 12. 'smaller elevations' Relative or absolute, or both?

p. 12, l. 22 and 25. The mentioning of the chemical formula in the parentheses is not necessary as al-ready introduced earlier on p. 4.

p. 13, l. 14. 'Mixing ratios in Taiwan . . . ' for which years?

p. 16, l. 34: Can you rule out that EDGAR has taken some emission results from Laube et al., 2012 to feed into their calculations and output? There are rumors that EDGAR is not always strictly taking in-ventory emission estimates. It might be worth to ask. Similar question for EDGAR and Ivy results (n-C7F16).

p. 17, l. 17. Please rephrase second part of that sentence, it is difficult to understand. Also the com-ment in parentheses (our calibration changed by 11%) appears odd, it does not seem to be the cali-bration that changed much, but the measured air samples changed a lot due to the fact that separa-tion is now possible.

p. 18, l. 1: eliminate one of the 'with'

p. 18, l. 28. How can cumulative emissions represent a percentage increase? Please rephrase. Publish the measurement results in the supplement, preferably, in tables/files that are easy to down-load and use for future users. Clearly state, which calibration scale the individual data sets are report-ed on.

Table 1, footnote a). State somehow that the assumption made on lifetime is one made in the present work, and not in Myhre et al.

Table 4: What are the analytical precisions of the calibrations exactly? Is there a reason they are given as ranges, and not as mean value? It is difficult to understand for several reason, one because there is not a clear mentioning of how many primary calibration standards are made, if and how these are propagated (to what) and how many working standards are used. Also, mixing ratios in the primary calibration mixtures should be given. Explain 'accuracy for CFC-11' more in detail. Why isn't this simp-ly a deviation (ratio) between the UEA primary calibration CFC-11 and that of NOAA? Again, why ranges? Also, ranges are given in an inconsistent way, sometimes from a small number to a large number (-1.4 to 2.8) and sometimes from a large number to a small number (-1.78 to -5.47). Footnote a) I don't understand the second part of the sentence (by calibrating pure compounds). I don't understand footnote b), please rephrase.

Figure 2: Contribution of what? Spelling mistake in legend (Cenral China)

Figure 5. It is very difficult to quickly get a good overview on this plot. The comment on the second-ary axis does not help much to directly understand, which of the two axes belongs to which com-pound. Suggest to extend '(blue circles)' by '(blue circles, left axis)', and correspondingly for '(green circles)', and then delete the sentence 'Note that . . ..'. Also, it would help a lot of the upper three sym-bols and descriptions would correspond to the upper part of the plot, i.e. suggest to put the three i-C6F14 legend entries on top (the current reversed way is the part that confuses most in this plot) (see Fig 4 where this is done correctly). Or split the legend in two. It would help to color-code the axis number and labels according to the plot colors. Lastly, it would also

[Figure]

help if the left axis numbers stopped at the maximum values of the concentrations, i.e. at around 0.08 ppt. The sentence 'Data pri-or to 2010 . . . . ' is incorrect, there are open circles up to about 2013. There is either no light blue or no dark blue shaded area visible on this plot. The symbol plot colors are difficult to distinguish and in some cases of similar tone for the two different compounds. The legend symbol size should be made larger (on a printed copy, they can hardly be distinguished in their colors). Suggest to set the left x axis limit somewhat prior to 1978 to show the 1978 results in full (there seems to be one green circle at 1978. No blue circle?). Space required before '[ppt]' for n-C6F14 for both main and inset axis. Some of these comments apply to some of the other figures.

Fig. 7: Add the Muhle et al. 2019 emission results to this graph. Fig 7 – 9: Colors are difficult to distinguish. Use methods to better distinguish the various lines. E.g. in Fig. 7, the Oram et al. results aren't visible for most of the record.

References: p. 24, l. 18: capitalize 'US'.

Fig. S1. Section reference missing (currently ??).

Fig. S2. Line 2. 'samples collected after 2005'? or 'Samples measured after 2005'? Why? There is a large difference between the ion ratios for the calibration standard and the tank samples, why? This large difference implies that there is still some kind of a co-elution on the column used in the present work. Perhaps an integration problem? Mention here or elsewhere in the text, what mixing ratios the 'new' calibration standard has, and perhaps here, what the differences in peak sizes are typically for the CGAA samples and for the cal standard.. Also, for some compounds, the ion ratio of the 'old' Laube 2012 calibration scales could be added to here.

---

## Referee Comment (RC2) · Anonymous Referee #2 · 4 Dec 2019

This paper presents measurements of PFCs at Cape Grim, Australia between 1978-2017, supplemented by measurements from Tacolneston, UK 2015-2017 and campaign-based sampling in Taiwan between 2013 and 2016. Long-term trends in mixing ratio and emissions are provided by the Cape Grim data, with the Tacolneston and Taiwan data providing information about emissions in their surrounding regions and the interhemispheric gradient. Although measurements of some of these PFCs at Cape Grim were presented in Laube et al., (2012), this new study is valuable because it has improved the calibration scale, allowed separation of some isomers, and extended the trends in time. I recommend publication after consideration of the following minor comments:

General comments:

1. Near where the isomers are first mentioned in the Introduction, it might be worth providing a paragraph that has a little more background on isomers (e.g., definition, explain the nomenclature n-isomer, i-isomer), and bringing this together with explanation of the significance of measuring the different isomers. The potential for distinguishing different source types is mentioned on page 4, line 23 and the different radiative efficiency is mentioned on page 4, line 9, and there is some discussion of the implications of separation of isomers from a measurement point of view elsewhere, but these points are interspersed with lots of other details, risking losing the overall significance of the work. A paragraph that discusses the significance of isomers in general, before going on with the details of this study, would be beneficial.

2. A clearer description of uncertainties is needed in Section 2.5. E.g., page 8, line 31 - what is the "averaged model-fit uncertainty of the Cape Grim trend" and how is it calculated? page 9, line 9 - "After these uncertainties were calculated" - uncertainties in what, PFC trends? Page 9, line 9 - "the model was re-run" - the inversion for emissions? I don't really understand this paragraph (lines 9-14). I do understand why you would want trend uncertainties without calibration uncertainty, it is the details of the description that I believe should be improved.

3. I am confused about the conclusion regarding the interhemispheric gradient of n-C6F14 and i-C6F14 (last sentence of section 3.1.3). Around page 13, line 7, I thought the authors were saying that the observations were close to or slightly exceeding a ratio of N:S mixing ratio the same as c-C4F8 (1.05), but that the measurement uncertainties were too large to discern the interhemispheric gradient. The conclusion at page 13, line 20 says that the NH sources are not as substantial - is this referring to NH sources relative to SH sources, and therefore influencing the interhemispheric gradient? We know that the recent total emissions for c-C4F8 are around 10 times the emissions of n-C6F14 and i-C6F14, so it is perhaps not surprising that the interhemispheric gradient will be small in an absolute sense, but that doesn't tell us about the relative sense. But I don't think we can conclude anything about the N-S distribution of emissions due to

the N-S mixing ratio difference expected for typical N-S emissions distribution, relative to the measurement uncertainty.

Specific comments:

page 3, line 4 - "As a result" - of what? Provide a better link to the previous sentence.

page 3, line 13 - "Other sources include ..." is this still talking about the low mass PFCs? Maybe "Other sources of PFCs include..."

page 10, line 16 - perhaps replace 'sources' by 'pieces' or 'lines', and leave the word 'sources' to describe emissions to the atmosphere. Same on line 19.

page 10, line 21 - "This is slightly less..." - what is? The 2017 value is slightly less?

page 12, line 2 - specify whether this is the interhemispheric ratio of emissions or mixing ratio

page 12, line 22 - "The trends are somewhat similar" - to what? Each other? Another PFC?

page 12, line 23 - The fastest increase occurred in the mid to late 1990s.

page 13, line 4 - I would add the years, i.e. "Mixing ratios in Taiwan in 2015 and 2016 are again ..."

page 15, line 21 - change to "for n-C4F10 (Fig. 8) and n-C5F12 (emissions $< 5.3 \times 10^{-5}$ Gg/yr)." as only emissions for n-C4F10 are shown in Fig 8. I know this is specified in the caption, but simpler for the reader to also specify here.

page 16, line 4 - Add "Cape Grim" as follows: "However the Cape Grim data in this time period"

page 16, line 15 - "although within the uncertainties of the early data set this is not a significant difference" - isn't it the sparsity of the data (i.e. timing of the increase depends on one sample, around 1996) rather than the data uncertainties as indicated

in the figure that mean we can't be conclusive about the timing of each increase? Is there a clue to the timing (of the n-C6F14 increase, at least) from the Ivy measurements of the Cape Grim air archive, as these are more dense in time?

page 16, line 16 - What does 'It' refer to? i-C6F14 emissions, presumably, but please specify.

page 16, line 29 - 'peak emission rates have increased' - please be clear what you are comparing here (n-C6F14 alone and the sum n-C6F14 + i-C6F14?)

page 16, line 33 - what is being compared to EDGAR emissions here? Are you comparing the black dashes with the red dots? Do they agree exceedingly well?

page 18, line 1 and Table S4 - is this talking about the correlation with CO described at the end of page 9? Or just footprints? Please explain what this is.

Fig 5 - I found this figure confusing because it is combined, and I think it would be less confusing if it were split into two panels. It is the use of different axes I found most confusing - it gives the wrong impression of the relative magnitude of the two isomers. I had to keep checking the caption/legend to remember which axis to use, and which isomer was which, as I came back to the plot a number of times as I read through the paper. The gridlines (although faint) don't match the right axis. The one advantage of combining the plots is that it is easy to compare the timing of the increase around 1995 for the two isomers, but this could also be done with one plot above the other. If the plot is to remain combined, I suggest plotting the different isomers on the same y-axis. Alternatively, if the different axis ranges are to be maintained then color the axis tick labels and text blue and green to correspond to the colors of the symbols and shaded areas of each isomer, as in Fig 8. You could also put 'i-C6F14' in blue text near the i-C6F14 curve, and similar for n-C6F14 (green text). I think would still prefer it split into 2 panels.

Fig 5 - as the i-C6F14 data only begin around 1987, should the figure only show the

red line and blue band from then? Otherwise it gives the impression that mixing ratio is known before 1987, but it isn't here.

Fig 7 - could color the right y-axis text and the EDGAR data as a color other than black, to emphasise that it is on the secondary axis.

Fig 9 - it is hard to distinguish the red and magenta dotted lines on my printed copy - I suggest using a more different color or line type.

Figures - it might be good to combine the mixing ratio and emissions figures for each PFC, for example combine Figs 3 and 7 as two panels in the same figure, Figs 4 and 8 etc. I found I wanted to look at the mixing ratio plot when I was reading about the emissions and looking at the emissions plot, and combining them would remove the need for flicking back and forth. The text could still remain in the same order, just describe the top (mixing ratio) panels first and come back to the lower (emissions) panels.

The measurements should be made available in a data archive or as downloadable tables in the Supplement.

Technical corrections:

page 12, line 18 - Hemisohere

page 12, line 19 - add 'is' after 'This'

page 16, line 26 - add 's' to 'emissions'

page 18, line 1 - with with

page 20, line 13 - exist, not exists

Supplement page 2, line 3 - Section ??

---

## Referee Comment (RC3) · Anonymous Referee #3 · 24 Dec 2019

This manuscript presents measurements of the atmospheric mixing ratio of six per-fluorocarbons (PFCs) obtained from a background site (Cape Grim) in SH for 1978-2018, and from two regional sites – in Tacolneston, UK for 2015-2017 and in Taiwan for four campaign periods between 2013 and 2016. It includes updates of previous measurements shown in Laube et al., (2012) by separating the isomers of n-C6F14 and i-C6F14 and improving the corresponding calibration scale. Authors also provide global emission estimates for these six PFCs in comparison with inventory-based and previous observation-based emissions estimates. This paper contains significant addition to current observation database and thus is worthy of publication in Atmospheric Chemistry and Physics after the following discussions are considered.

1. P6 LN 14-16: Since the update of the calibration scale was suggested as one of

the key purposes for this article, the absolute calibration procedure needs to be over-viewed more in detail than in the current version, and thus readers don't have to search for and read through previous studies cited here.

2. P 7, LN 8: Clarify that the combined influence is <2.8%. Is it a relative difference be-tween the $n$-$C_4F_{10}$ background concentrations determined based on the two different scales?

3. P 7, LN 14: What does the ion ratio means? Is it a ratio of peak heights for two chromatograms of m/z=219 and 169? Or a ratio of peak areas?

4. P 7, LN 14: Fig. S2 was shown earlier than Fig. S1 in the text.

5. P 7, LN 25: the accuracy derived from the CFC-11 reference compound seems to imply only an accuracy of the dilution process, not "the accuracy of the calibrations" that stated in the text. Otherwise more clarification is needed.

6. P 7, LN 30: How was the uncertainty of 0.3% determined? Was it 1-sigma? Then how many data were analyzed (n=?)?

7. P 7, LN 31: How could the ranges (4.0-7.8%, and -5.5-2.8%) of calibration accuracy be determined? More detailed explanation should be given.

8. P 7, LN 34: Again, please describe how the overestimation by 20% and 11% could be determined?

9. P 8, LN 15-19: The emission distribution for all PFCs seems to be based on the global distribution of $C_6F_{14}$ emissions recorded in the EDGAR database. The figures 7-10 showed the individual EDGAR emission estimate for each PFCs. Then authors need to discuss about how different (or consistent) the emission modeling results would be if the EDGAR emission estimate for each compound were used, instead of the $C_6F_{14}$ emissions record.

10. P 9, LN 6-8: This statement conflicts with a previous comment in page 7 (lines 8-

10): "Due to the leak-tightness of the system and a lack of observed isomers, a revision of the calibration scales for c-C4F8 and n-C5F12 was deemed unnecessary. If within the calibration uncertainty, their re-calibration was not necessary, why an additional uncertainty besides the calibration uncertainty should be added for these compounds?

11. P 10, LN 6-11: Explain more in detail how a certain correlation between modeled CO and the PFC mixing ratio can represent which emission source is more associated with PFCs and what extent as well?

12. P 10, LN 28: Clarify where the number of 1.05 came from.

13. P 10, LN 29-32: Please compare the current c-C4F8 measurements in Tacolneston with those from Mace Head in Muhle et al. (2019) and discuss and justify the statement of "the lack of variability in the data suggests that Tacolneston is not in close proximity to any major sources of c-C4F8. When considering the simulated atmospheric trend from the emission model for Tacolneston, it can be noted that it overestimates the observed concentrations, despite the well-fitted Southern Hemispheric trend."

14. P 11, LN 6-8: Are they based on the current simulation? Otherwise, add the correspond references.

15. P 11, LN 30-31: Please discuss how authors could exclude a possibility that the found similarity in the mixing ratio trends between n-C4F10 and n-C5F12 might be due to the fact that they were determined using a same m/z (119).

16. P 11, LN 18: Much lower mixing ratio elevations were observed where?

17. P 12, LN 33 – P 13, LN 2: Please explain how author can confirm the ratio shift from 2003-2008 to 2013-2018 periods was statistically significant. How was the uncertainty of +/-0.01 determined? Are there any possibility that the ratio shift could be related with the re-calibration after 2010?

18. P 13, LN 10-11: For the observed high mixing ratio, authors need to examine their trajectories to argue an influence of a pollution plume.

19. P 18, LN 1: remove the first "with".

20. P 18, LN 10-17: Authors should describe much more in detail about the correlation analysis between PFCs mixing ratio versus CO mixing ratio derived CO source type, which was not given even in the Supplementary information. How can the CO mixing ratio be distinguished by the sources? The description and resulting discussion should be provided in the main text.

---

## Author Comment (AC1) · 21 Feb 2020

In this document, we address the comments of anonymous referees #1, #2, and #3 to our paper on Trends and Emissions of Six Perfluorocarbons in the Northern and Southern Hemisphere. Referees' comments are highlighted in blue (in the supplement of this response) and are numbered according to referee number, page of the comment, and comment number, the latter of which we attributed ourselves wherever original comments were unnumbered. For example, the 23rd comment made by Referee #2, which can be found on page C3, will be referred to as 2.3.23 (ref.page.comment). Our responses are inserted below each comment of the referees. General comments have been addressed within our responses to the specific comments of the referees. References made to page and line numbers in our responses refer to the page and line

numbers in the original documents of the paper, not the revised documents, unless stated otherwise.

Please be aware that all our responses can also be found in the uploaded supplement to our interactive comment, which maintains a clearer visual structure and overview.

Elise Droste

Karina E. Adcock, Matthew J. Ashfold, Charles Chou, Zoë Fleming, Paul J. Fraser, Lauren J. Gooch, Andrew J. Hind, Ray L. Langenfelds, Emma Leedham Elvidge, Norfazrin Mohd Hanif, Simon O'Doherty, David E. Oram, Chang-Feng Ou-Yang, Marios Panagi, Claire E. Reeves, William T. Sturges, and Johannes C. Laube

————————————————————————————————————

General Notes:

- Table 4 has been changed to table 3; table 3 is now table 4.

- Referees have been included into the Acknowledgements section

————————————————————————————————————

Response to Referee #1

General Comments:

NA

Specific Comments:

1.2.1 Abstract, p. 1, l. 5: Perhaps add 'analytically', to clarify what 'separation' is discussed here (analytically separated from their isomers : : :)

Done.

1.2.2 p. 2, l. 4: is 'metric' necessary (isn't ACP using SI units anyway?).

After considering this comment, we decided to keep "metric" for extra clarity on the unit of measurement.

1.2.3 p. 2, l. 6 'despite what?', what does 'this' reference to?

Original sentence: "Despite this, the sources of all PFCs covered in this work remain poorly constrained and reported emissions in global databases do not account for the abundances found in the atmosphere."

"this" referred to the continuing emissions of the PFCs. However, we have decided to update the sentence and remove "despite this", since it is unnecessary and the remainder of the sentence brings across the message we would like to make: "Sources of all PFCs covered in this work remain poorly constrained and reported emissions in global databases do not account for the abundances found in the atmosphere."

1.2.4 p. 2, l. 5: Does the two-thirds refer to the emissions of the last 8 years or to the emissions since 1978?

Original sentence: "... 23% of which has been emitted in the last eight years. Almost two-thirds of the CO2 equivalent emissions are attributable to c-C4F8, which ... "

The two-thirds refer to the emissions of the last 8 years. This has been clarified in the text in the following manner:

Revised sentence: "Almost two-thirds of the CO2 equivalent emissions in the last eight years are attributable to c-C4F8, which currently also has the highest emission rates that continue to grow"

1.2.5 p. 3, l. 18: is decofluorobutane. Spelling here different to abstract

We thank the referee for spotting these details.

Decofluorobutane has been changed to decafluorobutane

1.2.6 p. 3, l. 19: hexodecafluoroheptane. Spelling here different to abstract.

Done. Hexodecafluoroheptane has been changed to hexadecafluoroheptane

1.2.7 p. 3, l. 21: same issue for perfluoro-2-methylpentane. Suggest to recheck spelling of all chemicals and use consistently throughout.

Done. Perfluoro(2-methylpentane) has been changed to perfluoro-2-methylpentane

1.2.8 p. 3, l. 24. ': : : of these: : :' would strictly refer to c-C4F8 and n-C5F12, but I presume that the author means all compounds handled in this manuscript.

Correct. "of these" replaced with "all six compounds studied in this manuscript"

1.2.9 p. 3, l. 24. In Table 1, there is a compound with a GWP outside the range indicated here (i-C6F14).

Original range on p. 3 line 24: "range between 7820 (n-C7F16) and 9540 (n-C4F10)"

This has been updated to "range between 7370 (i-C6F14) and 9540 (n-C4F10)"

1.2.10 p. 4 objective 2: There is something fundamentally unclear here. 1) Why were new calibration scales needed? I understand that with new methods, isomers could now be separated. 2) However, were the old calibration scales not made with pure reagents, e.g pure n-C4F10, no i-C4F10? If so, then why weren't the old calibration reference standards used for the present work, and the air sample data corrected for the fact that now the two isomers are separated? 3) It appears that the old calibration reference standards would still be good, but the measurement of the air samples would need to be redone or the old ones corrected. 4) Was the same raw material used to produce the new scales. 5) How many cali-bration reference standards were produced — just one?

1) The reason that new calibration scales were needed is that, when the PFCs had been calibrated, the isomers had not been separated isomers yet. We were not able to keep the calibration standards used to construct the previous calibration scale and so these could not be re-used.

2) We were not able to confirm that the pure compounds from previous calibrations indeed consisted of one isomer only. We used a technical mixture for n-C7F16 in the previous calibration due to lack of availability of a more pure compound, which only contained about 85% n-isomer.

3) Calibration references cannot be kept in our lab. The only thing we keep are the secondary standards (working standards), which contain clean Northern Hemispheric air and therefore a mixture of PFC isomers.

4) The same material was used of nC6F14, nC4F10. It was confirmed for these reagents that they did not contain any detectable amount of any other isomer. The pure reagents for nC7F16 and iC6F14 were newly purchased at purities specified in the manuscript.

5) For each of the calibrated PFCs, three calibrations were done. For each of these calibrations, the pure compound was diluted to mixing ratios anywhere between 4 and 10 ppt. This information has now been included in an additional table in the supplement, following suggestions in comment 1.5.32.

The following paragraph has been added at the beginning of Section 2.3:

"The mixing ratios in the samples are determined based on a secondary calibration standard, which we refer to as the working standard. Our working standard consists of clean Northern Hemispheric air, which therefore contains all relevant PFCs. For many gases, the mixing ratios in the working standard have been calibrated by NOAA. As certified values do not exist for the relevant PFCs, we calibrate them at UEA using an independent calibration scale (see more details in the Supplement)."

We also added the following statement on line 1 on page 7:

"We used the same compounds as in Laube et al. (2012) for n-C4F10 and n-C6F14, for which we now confirm their isomeric purity. For i-C6F14 and n-C7F16 we used newly purchased pure compounds, which were also confirmed to be isomerically pure. In

addition, for n-C7F16 this represents a significant improvement to Laube et al. (2012), as the latter used a technical mixture with an 85% n-isomer content."

1.3.11 p. 5, l. 14 – 16. It seems that this was already explained in the Intro objective 1.

Correct. Lines 14-16 on p5 have been removed ("Air samples collected for the Cape Grim Air Archive and UEA represent well-mixed, clean, Southern Hemispheric air, as the sampled air masses originate from the long trajectories over the Southern Ocean. Hence, the trace gas abundances in these samples can be considered as mid-latitude Southern Hemispheric background levels.").

1.3.12 p. 6, l. 7: Were compounds used in this study also affected? If not, perhaps be more specific by saying ..'number of compounds not used in this study'.

We are not stating that CO2 affected the signal of any compound. We are explaining that the placement of an Ascarite trap removes the CO2 and that its removal means that none of the measurements are distorted. We have clarified the sentence by re-phrasing it like so:

Original sentence: "For compounds analysed on the CP-PLOT column, an Ascarite (NaOH-coated silica) trap is included before the magnesium perchlorate trap to remove CO2, which can distort or reduce the signal of a number of compounds."

Revised sentence: ". . . an Ascarite (NaOH-coated silica) is included before the magnesium perchlorate trap to remove CO2, which can otherwise distort or reduce the signal of a number of compounds, including those discussed in this study."

1.3.13 p. 7, second para. Comment here is similar to p. 4, objective 2.: 1) '..from the observe isomer' observed where – in the old primary calibration standards? 2) If there was i- C4F10 in the old standards, why wasn't it precisely quantified by measuring it vs the new cal standards, then allowing to separate the <2.8% difference into the two factors 'leak-tightness' and 'i-C4F10' influence. 3) Line 9. Lack of observed iso-mers where?

[Figure]

1) We have observed the i-C4F10 isomer in the working standard and atmospheric samples. We have now clarified this by modifying this sentence:

Original sentence: "...any bias from the observed i-C4F10 isomer, and to ..."

Rephrased sentence: "...any bias from the observed i-C4F10 isomer (a small side peak of which was observed in both the working standard and atmospheric samples), and to ..."

2) The i-C4F10 isomer can currently not be quantified yet with the current method, because, apart from having a very small signal, its main quantifying ion is not well separated from one of the quantifying ions for n-C4F10. Even if we were able to separate it completely, it would also have to be calibrated using a pure i-C4F10 reagent, which we were unable to acquire. Hence, we are unable to quantify what fraction of the 2.8% is due to leak-tightness and to the influence of i-C4F10. This statement has been included in the Supplementary Material document under a new section, named "Additional Calibration Details", and we refer the reader to the Supplement for more details in line 8 on page 7.

3) P. 7, line 9: Original sentence: "Due to the leak-tightness of the system and a lack of observed isomers, a revision of the calibration scales for c-C4F8 and n-C5F12 was deemed unnecessary."

Revised sentence: "Due to the leak-tightness of the system and a lack of observed isomers of c-C4F8 and n-C5F12 in the working standard and atmospheric samples, a revision of the calibration scales was deemed unnecessary for these two compounds."

1.3.14 P. 7 l. 12: 'The confidence in the complete separation of all isomers for C6F14 and C7F16 : : :' This sen-tence is in conflict with earlier mentioning of inabilities of separating all isomers of e.g. C7F16.

We are able to separate the two significant isomers of C6F14 (n and i) and C7F16 (n

and i), but the latter also has a third peak that is not well separated from the peak for the i-C7F16 isomer. In this line, we are referring to the complete separation of the i- and n- isomer of C6F14 and similarly for C7F16. In other words, we have confidence that the peak for i-C6F14, n-C6F14, and n-C7F16 indeed only represent the signal for i-C6F14, n-C6F14, and n-C7F16, respectively.

To clarify, we have rephrased the sentence on lines 26-27 on page 6:

Original sentence: "Additionally, these two smallest peaks are not well separated from each other and the number of possible isomers with very similar mass spectra is too high to allow for a high-confidence identification."

Revised sentence: "Additionally, these two smallest peaks are not well separated from each other (but are well separated from the main isomer peak) and the number of possible isomers with very similar mass spectra is too high to allow for a high-confidence identification."

1.3.15 p. 7, l. 19: ': : : did not differ significantly: : :' seems incorrect, According to Fig. S2 they differ very strongly and significantly and are rather concerning.

We disagree with the referee on this point. Given the error bars on the ion ratios for the Cape Grim samples and the uncertainty of the ion ratios of the calibrations, the difference is not significant. The sentence prior to the one this comment refers to also states that we can attribute the confidence "within the uncertainties". We have changed the sentence on line 19-20 on page 7 as follows:

Original sentence: "For i-C6F14, n-C6F14, and n-C7F16, the ion ratio in the Cape Grim samples did not differ significantly from the ion ratio in the calibrations (Fig. S2)."

Revised sentence: "For i-C6F14, n-C6F14, and n-C7F16, the ion ratio in the Cape Grim samples did not differ significantly from the ion ratio in the calibrations within 1 sigma measurement uncertainty (Fig. S1)."

1.3.16 p. 7, l. 27. The abbreviation does not seem to match the first letters of the full

name. We thank the referee for noticing this mistake. We have now corrected both the abbreviation and the name itself.

Original: "... CMD (Global Measurement Division) ..."

Revised: "... GMD (Global Monitoring Division)..."

1.3.17 p. 7, l. 30: 1) Could you provide a guess why there is a difference between the bootstrapped CFC-11 and that measured on the NOAA scale. 2) There is a lot of plural mentioning, but isn't it only one compari-son, that of the (single) working tank vs the primary calibration tanks?

1) The overall volume uncertainty of the sample loops that were filled with pure compounds during both dilution steps is 5%, as has been outlined in Laube et al., 2010. This is the likeliest and highest source of uncertainty in the entire calibration procedure. As has been shown in multiple previous papers (Laube et al., 2010; 2012; 2014;2016; Oram et al., 2012; Kloss et al., 2014), the overall calibration uncertainty is very likely about 7%. Our measurements fall well within that envelope. This information has now been included in a new section in the Supplement, named "Additional Calibration Details".

2) For each compound in this work, dilutions of the pure compound were created three times (on different days) and these were analysed (along with CFC-11) on the day that they were created. Each compound has thus been calibrated at three times in order to obtain calibrations uncertainties. This has been clarified in the text by adding the following sentence on line 34 on page 6:

"This calibration procedure was repeated three times for each compound and for a range of dilutions (Table S1)."

We have also included more detailed information on the mixing ratios the diluted compounds in an additional table in the Supplement (see comment 1.5.32), to which we now refer in the text.

[Figure]

1.3.18 p. 7, l. 31. How can an accuracy be negative (-5.5%).

This is a good point. It is strictly speaking indeed not the accuracy, but instead an offset of our calibrations compared to calibrations done by NOAA. We use this offset as a measure of the accuracy of our calibrations, as the NOAA CFC-11 scale is one of the most internationally recognised calibrations. The negative value therefore indicates that the lab calibrated mixing ratio determined for the reference compound is lower than the mixing ratio certified by NOAA. To make this clearer, we have amended lines 25-30 on page 7:

Original text: "To determine the accuracy of the calibrations, trichlorofluoromethane (CFC-11) was diluted along with the pure PFCs as a reference compound, as our working tank has been calibrated by the globally recognised CMD (Global Measurement Division) of the NOAA-ESRL (National Oceanic and Atmospheric Administration - Earth System Research Laboratory) for CFC-11. For all UEA calibrations, the average difference with values determined by NOAA show that the UEA calibrated concentrations for CFC-11 are consistently slightly lower by on average 4.2%+- 0.3%."

Revised text: "To obtain a measure of accuracy of the calibrations (as in Laube et al. 2010; 2012), trichlorofluoromethane (CFC-11) was diluted along with the pure PFCs as a reference compound, as our working tank has been calibrated by the globally recognised CMD (Global Measurement Division) of the NOAA-ESRL (National Oceanic and Atmospheric Administration - Earth System Research Laboratory) for CFC-11. For all UEA calibrations, the average offset with values determined by NOAA show that the UEA calibrated concentrations for CFC-11 are consistently slightly lower by on average 4.2%+- 0.3%."

Original text in caption of Table 4: "Mixing ratios determined in the working standard, precisions of the calibrations, and accuracies of CFC-11 calibrations compared to NOAA scales."

Revised text: "Mixing ratios determined in the working standard, analytical precisions

of the calibrations, and measures of accuracies of CFC-11 calibrations expressed as relative differences to NOAA scales ((UEA-NOAA)/UEA)."

"Accuracy for CFC-11 [%]" in header of Table 4 has been changed to "CFC-11 Difference to NOAA [%]"

1.3.19 p. 7, l. 31: It is incorrect to say that the improvement by the new calibration scale is significant. The improvement is max 2.8% as stated earlier. What improved a lot are the measurement results of the field samples, because i-C4F10 is now excluded from the n-C4F10 measurements. Again, this is mainly a correction of older air measurements, not an improvement of a scale.

We agree that the improvement is max. 2.8% for C4F10. However, we are not stating that there is a significant improvement of the C4F10 calibration scale. In the text referred to by the referee, we are specifically referring to the calibration scales of C6F14 and C7F16. We are also not stating that the improvement for these two PFCs is significant. Instead, we state that it is substantial, as indicated by the degree of overestimation of the mixing ratios for the n-isomer of C6F14 and C7F16. With regards to the improvement of the calibration scale, please also see our response to comment 1.2.10.

1.3.20 p. 7, l. 34: 1) How did you get to the 20% and 11%. Perhaps explain in more detail inthe supplement. 2) Labeling a calibration scale 'new' calls for problems down the road. It would be much clearer to give it a name.

1) The 20% and 11% is determined based on the relative difference in the concentration determined for the n-isomer of C6F14 and C7F16, respectively, in the working standard between the two (old and new) calibration scales. The old calibration scale was derived before the isomers were separated (i.e. when the signal in the working standard was calibrated against pure n-isomer when it actually consisted of multiple isomers) and the new calibration scale was derived when isomers in the working standard were analytically separated.

Instead of including this in the supplement, we decided to clarify this in the text (page 7, line 34) by inserting the following: "These levels of improvement in the UEA2018 calibration scale are determined based on the relative difference in concentration determined for the respective n-isomers in the working standard between using the UEA2010 and the UEA2018 calibration scales."

2) We agree with the referee that this is a good idea. We have done so by modifying p. 6 line 31, in which we now give the old calibration and new calibration scale the following labels: UEA2010 and UEA 2018, respectively. Please also see the given labels used in the response to the previous comment.

Original sentence: "The previously reported calibration scales (Laube et al., 2012) were revised to accommodate the separated isomers."

Revised sentence: "The previously reported calibration scales (Laube et al., 2012) (now referred to as UEA2010) were revised to accommodate the separated isomers. The new calibration scale is referred to as UEA2018"

1.4.21 p. 8, l. 4: 'horizontal'. Shouldn't this be 'vertical'? To avoid confusion, we have removed "horizontal" and replaced it with "atmospheric".

Original sentence: "The model domain consists of 24 equal-area, zonally averaged latitudinal bands, which each have twelve horizontal layers between the surface and the maximum altitude represented in the model."

Revised sentence: "The model domain consists of 24 equal-area, zonally averaged latitudinal bands, which each have twelve atmospheric layers between the surface and the maximum altitude represented in the model."

1.4.22 p. 8, l. 7: Why would a 'not fully represented' atmospheric layer justify leaving it out, there is a good part of the stratosphere in the model.

PFCs are only destroyed by photolytic processes in the mesosphere, which is not represented in the model. Since there are no sources in the upper part of the model

domain, there is no gain in including the stratosphere for the purposes of determining global emissions. It has been recently been pointed out by Ray et al., Nature Geoscience, 2020, that the stratosphere can indeed have an impact on tropospheric long-term mixing ratio trends. However, this is only the case for species that have their main sink region in the stratosphere and can also be measured with very good precisions of <0.3%. Neither is the case for any of the PFC species measured in this work.

While responding to this comment, we noticed that we had mistakenly stated that photolytic loss of PFCs also occurred in the stratosphere. We have now corrected this sentence. See below.

Original sentence: "Photolytic loss would only be relevant at stratospheric and mesospheric altitudes (Morris et al., 1995), which are not fully represented in the model domain."

Revised sentence: "Photolytic loss would only be relevant at mesospheric altitudes (Morris et al., 1995), which are not represented in the model domain."

1.4.23 p. 10, l. 10: 1) Provide some more intercomparison for the c-C4F8 mixing ratio of this study and that of Muhle et al., 2019, either here or in the supplement. 2) Add the Muhle et al. CGAA mixing ratio to the present results. Are the deviations a constant ratio or are there potential nonlinearity issues. 3) Can you derive a calibration scale factor between the UEA and the SIO scales, perhaps based on the CGAA re-sults, or other ways of intercomparisons (Tacolneston?). A well-derived conversion factor is extremely useful e.g. for users of the many data sets presented here and in Muhle et al.

1) A comparison has already been done in the Muhle et al., 2019 paper, up until 2010 in the time series. Since the focus of our paper is on updating the time series, we have now included the following text on page 10 line 20-26, quantifying the differences between the results in Muhle et al. and in our work:

Original paragraph: "The c-C4F8 mixing ratio in the Southern Hemisphere increased from 0.31 ppt in 1978 to 1.52 ppt by the end of 2017 (Fig. q3). This is slightly less than the mixing ratios of _1.6 ppt reported in the recent work of MuÌĹhle et al. (2019) with the difference most likely being due to the two independent absolute calibration scales. As reported previously by Oram et al. (2012), the mixing ratios seemed to stabilise in the 1990s, but picked up again in the 2000s. The extended time series since 2008 indicate a continuing increasing trend similar to the one pre-1990, totalling in a 27 % enhancement in the last eight years (2010-2018). In fact, atmospheric c-C4F8 abundances show accelerating growth in recent years, potential reasons for which are explored in Section 3.2.1."

Revised paragraph: "The c-C4F8 mixing ratio in the Southern Hemisphere increased from 0.31 ppt in 1978 to 1.52 ppt by the end of 2017 (Fig. q3). As reported previously by Oram et al. (2012), the mixing ratios seemed to stabilise in the 1990s, but picked up again in the 2000s. The extended time series since 2008 indicate a continuing increasing trend similar to the one pre-1990, totalling in a 27 % enhancement in the last eight years (2010-2018). In fact, atmospheric c-C4F8 abundances show accelerating growth in recent years, potential reasons for which are explored in Section 3.2.1. Muhle et al. (2019) reported a slightly higher mixing ratio of 1.61 ppt at the end of 2017 in the high latitude Southern Hemisphere. Since Muhle et al. (2019) already compared their Southern Hemisphere time series to ours until 2008 (published in Oram et al., 2012), we here focus our comparison on the extended part of the record, i.e. from 2008 until 2017. The average difference of annual mixing ratios for this part of the time series is 0.09 +- 0.01 ppt (1-sigma). This difference, which is equivalent to an average of 7%, is to some degree likely due to the two different independent absolute calibration scales. However, since the difference is clearly not constant over time (see Fig. 3A), which has also been discussed in Muhle et al. (2019), no simple conversion factor can be derived for the two data sets."

Similarly, we have included more information on the comparison of the emission esti-

mates for c-C4F8 between the two papers on page 14 line 15-17.

Original text: "Annual emission are now approaching rates of 2.0 Gg yr-1, which are, within uncertainties, comparable to rates determined for the mid-1980s. This compares quite well with the estimated global emissions of 2.2 Gg yr-1 in 2017 by the recent work of MuÌĹhle et al. (2019)."

Revised text: "Annual emission are now approaching rates of 2.0 Gg yr-1, which are, within uncertainties, comparable to rates determined for the mid-1980s. This compares quite well with the estimated global emissions of 2.2 Gg yr-1 in 2017 by the recent work of MuÌĹhle et al. (2019). The average difference between Muhle et al.'s (2019) and our emission estimates for 2008-2017 is 0.17 +_0.09 Gg yr-1 (1-sigma). However, the two data sets agree within the uncertainty ranges. "

2) Done.

3) Please see our response to point 1.

1.4.24 p. 12, l. 12. 'smaller elevations' Relative or absolute, or both?

Original sentence: "Third of all, smaller Northern Hemispheric sources of n-C4F10 and n-C5F12 compared to c-C4F8 are reflected by the smaller elevations of their mixing ratios measured in Taiwan, despite their enhancement compared to Cape Grim measurements"

Rephrased to "Third of all, smaller Northern Hemispheric sources of n-C4F10 and n-C5F12 compared to c-C4F8 are reflected by the smaller elevations of their mixing ratios measured in Taiwan relative to those of c-C4F8, despite their enhancement compared to Cape Grim measurements".

1.4.25 p. 12, l. 22 and 25. The mentioning of the chemical formula in the parentheses is not necessary as al-ready introduced earlier on p. 4. The chemical formulas in lines 22 and 25 on page 12 have been removed.

1.4.26 p. 13, l. 14. 'Mixing ratios in Taiwan : : : ' for which years?

We are here referring to the mixing ratios measured in Taiwan between 2015-2016. The text has been updated: "Mixing ratios in Taiwan in 2015 and 2016 . . ."

1.4.27 p. 16, l. 34: Can you rule out that EDGAR has taken some emission results from Laube et al., 2012 to feed into their calculations and output? There are rumors that EDGAR is not always strictly taking in-ventory emission estimates. It might be worth to ask. Similar question for EDGAR and Ivy results (n-C7F16).

The EDGAR emissions and the differences between the EDGAR emissions and top-down emission estimates have not notably changed since the publication of the data in Laube et al., 2012, and Ivy et al., 2012. We therefore have no reason to believe such rumours.

1.4.28 p. 17, l. 17. Please rephrase second part of that sentence, it is difficult to under-stand. Also the com-ment in parentheses (our calibration changed by 11%) appears odd, it does not seem to be the cali-bration that changed much, but the measured air samples changed a lot due to the fact that separa-tion is now possible.

Original sentence: "However, to assume that the measured C7F16 in previous work only consists of n-C7F16 would overestimate n-C7F16 mixing ratios by an unknown amount (our calibration changed by 11 %, see Section 2.3), because even though previous work diluted high purity compounds in calibration work, the amount of likely unseparated isomers present in their working standard tanks and samples remains unknown."

Revised sentence: "However, to assume that the measured signal for C7F16 in pre-vious work only consists of the n-C7F16 isomer would overestimate n-C7F16 mixing ratios by an unknown amount (see Section 2.3), because the signal for C7F16 in the working standard tanks and in the air samples is not exclusive to the n-isomer, as isomers were not separated in the earlier studies."

We have removed the mentioning of the 11%, because it has been explained in Section 2.3 and is clarified according to comment 1.3.20. The second part of this comment (1.4.28) has also already been addressed in our response to comment 1.2.10.

1.4.29 p. 18, l. 1: eliminate one of the 'with'

Done.

1.5.30 p. 18, l. 28. 1) How can cumulative emissions represent a percentage increase? Please rephrase. 2) Publish the measurement results in the supplement, preferably, in tables/files that are easy to down-load and use for future users. 3) Clearly state, which calibration scale the individual data sets are report-ed on.

1) The cumulative global $CO_2$ equivalent of the PFCs discussed in the work are 23% higher in 2017 compared to 2010. We clarify it by rephrasing the sentence in question:

Original sentence: "This represents an increase of 23 % between the beginning of 2010 and the end of 2017."

Revised sentence: "This represents an increase in the total global cumulative $CO_2$ equivalent of 23% between the beginning of 2010 and the end of 2017. "

2) All measurement results are made publicly available through a DOI, in accordance with ACP guidelines. We have included a statement in the data availability section of the paper: "The measurement and modelling data presented in this work have been made available on Zenodo and can be publicly accessed using DOI 10.5281/zen-odo.3519317, or by contacting e.droste@uea.ac.uk."

3) The methods sections covers that n-C4F10, i-C6F14, n-C6F14, and n-C7F16 in this work have been re-calibrated in order to account for the separation of the isomers and that thus these compounds have a new calibration scale. It has also been stated that the re-calibration of c-C4F8 and n-C5F12 was deemed unnecessary, for reasons explained in the paper. The calibrations used for these compounds in this paper are thus the same as in previous work produced by our lab. We feel that the methods

section makes this sufficiently clear. In response to an earlier comment, we have given the calibrations scales a clear label, i.e. UEA2010 and UEA2018.

1.5.31 Table 1, footnote a). State somehow that the assumption made on lifetime is one made in the present work, and not in Myhre et al.

Done.

"The lifetime for i-C6F14 in this work is here assumed to be the same as . . ."

1.5.32 Table 4: 1) What are the analytical precisions of the calibrations exactly? 2) Is there a reason they are given as ranges, and not as mean value? It is difficult to understand for several reason, one because there is not a clear mentioning of how many primary calibration standards are made, if and how these are propagated (to what) and how many working standards are used. 3) Also, mixing ratios in the primary calibration mixtures should be given. 4) Explain 'accuracy for CFC-11' more in detail. Why isn't this simp-ly a deviation (ratio) between the UEA primary calibration CFC-11 and that of NOAA? 5) Again, why ranges? 6) Also, ranges are given in an inconsistent way, sometimes from a small number to a large number (-1.4 to 2.8) and sometimes from a large number to a small number (-1.78 to -5.47). 7) Footnote a) I don't understand the second part of the sentence (by calibrating pure compounds). 8) I don't understand footnote b), please rephrase.

1) The analytical precision of the calibration refers to the 1-sigma standard deviation of the same working standard that was run several times before and after each calibration run in the same manner as the analytical precision for the samples is determined. As each sample was calibrated three times, the range for the precisions are given in this table. We have updated Table 4 by adding the average analytical precisions for the calibrations and keeping the ranges in brackets.

The following line has been added on line 25 on page 7: "The latter is the relative 1-sigma standard deviation of the compound signal in multiple working standard runs

analysed on the day of each calibration."

The caption of Table 4 has been updated:

Original caption: "Mixing ratios determined in the working standard, precisions of the calibrations, and accuracies of CFC-11 calibrations compared to NOAA scales. For details on the calibration of c-C4F8 and n-C5F12, consult Oram et al. (2012) and Laube et al. (2012)"

Revised caption: "Mixing ratios determined in the working standard, average analytical precisions of the calibrations, and measures of accuracies of CFC-11 calibrations expressed as the average relative differences compared to NOAA scales ((UEA-NOAA)/UEA). The analytical precision is the relative 1 sigma standard deviation of the compound signal in multiple working standard runs analysed on the day of each calibration. The ranges of analytical precisions and differences to NOAA are shown here for each compound in brackets and include all three calibrations done in this work for that compound. For details on the calibration of c-C4F8 and n-C5F12, consult Oram et al. (2012) and Laube et al. (2012)."

2) By addressing comment 1.3.17, we have already added the information on how many calibrations were done per compound. In the text (line 25, page 7) we do indeed write that we are reporting average precisions in Table 4. We have decided that we will show the average precisions in the table, along with the range in brackets.

3) We have created a table (new Table S1) with this information and have appended it into a new section within the Supplement, named "Additional Calibration Details". Please see comment 1.3.17.

4) Please see the response to comment 1.3.18.

5) It was chosen to show ranges in order to stay consistent with the accuracy reported in Laube et al. 2012, which is also done in ranges. However, we agree that it would be clearer to show the average offsets to NOAA as well and have added them in Table 4.

Caption has been edited to be consistent with the changes to the table (see point 1 of this comment).

6) Ranges changed to be consistent (smaller to larger).

7) Original sentence: "Mixing ratios determined in working standard by calibrating pure compounds."

Rephrased sentence: "New calibration (UEA2018)."

8) As described in the text, some PFCs were not re-calibrated for the working standard used in this study as they do not have isomers that significantly affect the signal of the peak of the isomer in question.

Original sentence: "Mixing ratios determined in working standard by converting calibrated mixing ratios in the previous working standard to current working standard."

Revised sentence: "Old calibration (UEA2010)"

1.5.33 Figure 2: 1) Contribution of what? 2) Spelling mistake in legend (Cenral China)

1) Original sentence: "Regions for which the contribution to the footprint simulated by the NAME model is quantified."

Rephrased to: "Regions used in the NAME model to quantify the simulated contributions to particle densities in Taiwan."

2) The typo in legend has been corrected. Double check size of figure after recompiling file.

1.5.34 Figure 5. It is very difficult to quickly get a good overview on this plot. The comment on the second-ary axis does not help much to directly understand, which of the two axes belongs to which com-pound. Suggest to extend '(blue circles)' by '(blue circles, left axis)', and correspondingly for '(green circles)', and then delete the sentence 'Note that : : :.'. Also, it would help a lot of the upper three sym-bols and

descriptions would correspond to the upper part of the plot, i.e. suggest to put the three i-C6F14 legend entries on top (the current reversed way is the part that confuses most in this plot) (see Fig 4 where this is done correctly). Or split the legend in two. It would help to color-code the axis number and labels according to the plot colors. Lastly, it would alsohelp if the left axis numbers stopped at the maximum values of the concentrations, i.e. at around 0.08 ppt. The sentence 'Data pri-or to 2010 : : :. ' is incorrect, there are open circles up to about 2013. There is either no light blue or no dark blue shaded area visible on this plot. The symbol plot colors are difficult to distinguish and in some cases of similar tone for the two different compounds. The legend symbol size should be made larger (on a printed copy, they can hardly be distinguished in their colors). Suggest to set the left x axis limit somewhat prior to 1978 to show the 1978 results in full (there seems to be one green circle at 1978. No blue circle?). Space required before '[ppt]' for n-C6F14 for both main and inset axis. Some of these comments apply to some of the other figures.

We thank the reviewer for these useful suggestions and have implemented all of them apart from the following:

- Changing the order of the symbols in the legend. We have instead followed the alternative suggestion to split the legend.

- Limiting the (originally) left axis scale to the maximum values of the concentrations. The scales of the y axes are chosen in a way that the curves will not overlap in the figure.

- Changing the symbol plot colours. The colours in all figures have been carefully chosen as to be clearly distinguishable, by colour-blind people as well as on a printed grayscale copy.

- Shifting the left x axis limit. Our time series starts in 1978. We would like to keep consistency in all our graphs.

We recognised that the plot was difficult to quickly interpret, which is why we have also made the following additional changes in order to improve it:

- Switched the y axes so that the upper curve (now n-C6F14) is plotted on the left y axis, i.e. the one that it is closest to.

- Switched the colour coding of the two isomers to match the change above.

- Slightly enlarged the size of all markers in the plots to improve visibility.

- Removed the gridlines in response comment 2.4.19.

- We made changes to the caption in order to make it consistent with the changes in the plot (as outlined above). These also include the comments made in 1.5.34.

Original caption: "A) Mixing ratios at Cape Grim of i-C6F14 (blue circles) and n-C6F14 (green circles) between 1978 and 2018. Note that data for n-C6F14 are plotted on a secondary axis. Data prior to 2010 are shown as empty symbols (Laube et al., 2012), while data from samples collected after 2010 are shown as filled symbols to illustrate the part of the time series that is extended in the current work. Note that the data from Laube et al. have been converted to a new and improved calibration scale. The atmospheric trends simulated by the model are represented by the red line for i-C6F14 and by magenta line for n-C6F14. Total uncertainties (light blue for i-C6F14, light green for n-C6F14) and trend uncertainties (dark blue for i-C6F14, dark green for n-C6F14) are indicated with the shaded areas along the trend line. B) i-C6F14 mixing ratios after 2010 for Cape Grim (blue circles), for Tacolneston (orange diamonds), and Taiwan (light blue squares); n-C6F14 mixing ratios after 2010 for Cape Grim (green circles), for Tacolneston (light green diamonds), and Taiwan (magenta squares). The red and magenta lines indicate the modelled Southern Hemisphere baseline trend at Cape Grim for i-C6F14 and n-C6F14, respectively; while the dashed red and magenta lines indicate the modelled trend for mixing ratios at Tacolneston for i-C6F14 and n-C6F14, respectively. Total and trend uncertainties are indicated with the shaded grey

areas along the Tacolneston dashed-trend lines for both i-C6F14 and n-C6F14."

Revised caption: "A) Mixing ratios at Cape Grim of n-C6F14 (blue circles, left axis) and i-C6F14 (green circles, right axis) between 1978 and 2018. Data prior to 2010 are shown as empty symbols (Laube et al., 2012), while data from samples collected after 2010 are shown as filled symbols to illustrate the part of the time series that is extended in the current work. Note that the data from Laube et al. have been converted to a new and improved calibration scale. The atmospheric trends simulated by the model are represented by the red line for n-C6F14 and by magenta line for i-C6F14. Total uncertainties (light blue for n-C6F14, light green for i-C6F14) and trend uncertainties (dark blue for n-C6F14, dark green for i-C6F14) are indicated with the shaded areas along the trend line. Note that there is only a very small difference between the trend and total uncertainties and thus the latter is difficult to distinguish. B) n-C6F14 mixing ratios after 2010 (left axis) for Cape Grim (blue circles), for Tacolneston (orange diamonds), and Taiwan (light blue squares); i-C6F14 mixing ratios after 2010 (right axis) for Cape Grim (green circles), for Tacolneston (light green diamonds), and Taiwan (magenta squares). The red and magenta lines indicate the modelled Southern Hemisphere baseline trend at Cape Grim for n-C6F14 and i-C6F14, respectively; while the dashed red and magenta lines indicate the modelled trend for mixing ratios at Tacolneston for n-C6F14 and i-C6F14, respectively. Total and trend uncertainties are indicated with the shaded grey areas along the Tacolneston dashed-trend lines for both n-C6F14 and i-C6F14."

1.6.35 1) Fig. 7: Add the Muhle et al. 2019 emission results to this graph. 2) Fig 7 – 9: Colors are difficult to distinguish. Use methods to better distinguish the various lines. E.g. in Fig. 7, the Oram et al. results aren't visible for most of the record.

1) Done.

2) Done. The emission data from Laube et al. is now plotted in the same colour as the emission data from the current work (although the different line styles makes clear

distinctions between them). The emission data from Ivy et al. for C5F12 in figure 8 has been plotted in a different colour to make a clear distinction with the plotted data from Ivy et al for C4F10. The line styles have also been changed to make the distinction clearer. For Figure 9 specifically, we have changed the line style and colour for the curve illustrating the sum of emission estimates of the two $C_6F_{14}$ isomers.

Inspired by the suggestions made in comment 1.6.34 for the trend plots, we have made minor changes to the legends in the emission plots: we split the legend between symbols belonging to two different PFCs whenever we plot the emission estimates for more than one PFC in the same graph.

Captions of all emission plot figures have been updated accordingly. We hope that these changes have improved the emission plots.

1.6.36 References: p. 24, l. 18: capitalize 'US'.

Done.

1.6.37 Fig. S1. Section reference missing (currently ??).

Done.

1.6.38 1) Fig. S2. Line 2. 'samples collected after 2005'? or 'Samples measured after 2005'? Why? 2) There is a large difference between the ion ratios for the calibration standard and the tank samples, why? This large difference implies that there is still some kind of a co-elution on the column used in the present work. Perhaps an integration problem? 3) Mention here or elsewhere in the text, what mixing ratios the 'new' calibration standard has, and perhaps here, what the differences in peak sizes are typically for the CGAA samples and for the cal standard.. 4) Also, for some compounds, the ion ratio of the 'old' Laube 2012 calibration scales could be added to here.

1) Samples collected after 2005. This has been corrected.

2) This point has been addressed in our response to comment 1.3.15.

[Figure]

3) The mixing ratios determined for each PFC are already recorded in Table 4. The mixing ratios of the diluted pure compounds used in the calibration procedures are included into a separate table (Table S1) in the supplementary material. As the peak sizes in the CGAA samples substantially change over time, there is not "typical" difference with the working standard. We would be happy to provide these technical details on request, but this information is not relevant for any of the scientific messages in this paper.

4) The purpose of this particular graph is to show that the ion ratio in the calibration standards and the Cape Grim samples are consistent and not changing over time. We feel that adding outdated ion ratios would only confuse the reader.
* * *
Responses to Referee #2

General Comments:

2.2.1 Near where the isomers are first mentioned in the Introduction, it might be worth providing a paragraph that has a little more background on isomers (e.g., definition, explain the nomenclature n-isomer, i-isomer), and bringing this together with explanation of the significance of measuring the different isomers. The potential for distinguishing different source types is mentioned on page 4, line 23 and the different radiative efficiency is mentioned on page 4, line 9, and there is some discussion of the implications of separation of isomers from a measurement point of view elsewhere, but these points are interspersed with lots of other details, risking losing the overall significance of the work. A paragraph that discusses the significance of isomers in general, before going on with the details of this study, would be beneficial.

We agree with the referee that including more introductory details would be beneficial to the paper. We have added the following to the introduction (page 3 line 20):

Previous line:  Previously  reported  PFC  trends  in  the  atmosphere  include  CF4,

C2F6, octafluoropropane (C3F8), cyclic-octafluorobutane (c-C4F8), decofluorobutane (C4F10,) dodecafluoropentane (C5F12), tetradecafluorohexane (C6F14), hexodecafluoroheptane (C7F16), and octadecafluorooctane (C8F18) (Laube et al., 2012; Ivy et al., 2012a; Oram et al., 2012; MuÌĹhle et al., 2010; MuÌĹhle et al., 2019)."

New text added to the end of the paragraph: "No isomers of these PFC species have been reported in the atmosphere before. As the GWPs and atmospheric lifetimes can differ substantially between isomers (Bravo et al., 2010), it is important to ascertain the full analytical separation and quantification of such isomers."

2.2.2 A clearer description of uncertainties is needed in Section 2.5. 1) E.g., page 8, line 31 - what is the "averaged model-fit uncertainty of the Cape Grim trend" and how is it calculated? 2) page 9, line 9 - "After these uncertainties were calculated" - uncertainties in what, PFC trends? Page 9, line 9 - "the model was re-run" - the inversion for emissions? 3) I don't really understand this paragraph (lines 9-14). I do understand why you would want trend uncertainties without calibration uncertainty, it is the details of the description that I believe should be improved.

1) We have added more information on what the model-fit uncertainty entails and made a reference to an earlier section that covers the analytical uncertainty on page 8 line 32 : "The analytical uncertainty is described in Section 2.2. The model-fit uncertainty is the average relative difference between the observed mixing ratio and the simulated mixing ratio."

The average analytical uncertainty and average best-fit uncertainty have been added to (what is now) Table S1.

2) We have improved our explanation of how the emission uncertainties are determined:

Original sentence: "After these uncertainties were calculated, the model was re-run using the best fit of the observed mixing ratios adjusted by the uncertainties to estimate

the maximum and minimum emissions."

Rephrased: "The "trend uncertainty" and the "total uncertainty" for each PFC were used to determine the respective uncertainty bands around the observed mixing ratio trend. The model was then re-run to find the emissions that fitted the simulated maximum and minimum mixing ratios best. The result is the minimum and maximum emissions according to both types ("trend" and "total") of uncertainties."

3) Lines 9 to 10 have been rephrased in point 2. Lines 10-14 (see below) have been moved to the end of section 2.2, because we decided that even though these lines discuss the errors/uncertainty in the measurements, it makes more sense to keep this information with the section that actually discusses the measurements.

Lines 10-14: "Measurement errors (indicated by the error bars on the observational data points) consist of a combination of the 1 sigma standard deviations of the working standard and sample replicates on the same analysis day. For samples that have been analysed against the previous working standard, the uncertainties include the aforementioned internal conversion accounted for as the 1 sigma standard deviation of the peak ratio on inter-comparison days (n=8)."

The title of section 2.5 has subsequently been changed from "Uncertainties" to "Trend and Emission Uncertainties" to make this distinction clear.

2.2.3 I am confused about the conclusion regarding the interhemispheric gradient of n-C6F14 and i-C6F14 (last sentence of section 3.1.3). Around page 13, line 7, I thought the authors were saying that the observations were close to or slightly exceeding a ratio of N:S mixing ratio the same as c-C4F8 (1.05), but that the measurement uncertainties were too large to discern the interhemispheric gradient. The conclusion at page 13, line 20 says that the NH sources are not as substantial - is this referring to NH sources relative to SH sources, and therefore influencing the interhemispheric gradient? We know that the recent total emissions for c-C4F8 are around 10 times the emissions of n-C6F14 and i-C6F14, so it is perhaps not surprising that the interhemispheric gradient

will be small in an absolute sense, but that doesn't tell us about the relative sense. But I don't think we can conclude anything about the N-S distribution of emissions due to the N-S mixing ratio difference expected for typical N-S emissions distribution, relative to the measurement uncertainty.

We were indeed comparing the NH sources of the isomers of C6F14 to those for c-C4F8 in a relative sense. However, the comment made here is very valid and we have adjusted lines 18-20 on page 13:

Original sentence: "Even though a large range of mixing ratios is observed at Tacolneston and in Taiwan for both compounds, the lack of a discernible interhemispheric gradient (given the measurement uncertainties) suggests that the Northern Hemispheric sources are not as substantial as they are for, for example, c-C4F8."

Revised sentence: "Even though substantial variability of mixing ratios above the Cape Grim baseline for both C6F14 isomers is observed at Tacolneston and in Taiwan, the low absolute mixing ratios in both hemispheres combined with the measurement uncertainties do not allow for any further conclusions on the hemispheric distribution of their emissions."

Specific Comments:

2.3.4 page 3, line 4 - "As a result" - of what? Provide a better link to the previous sentence.

Original sentence: "... by their strong infra-red absorption properties and inertness. As a result, PFCs can have global warming potentials (GWPs) of ..."

Revised sentence: ""... by their strong infra-red absorption properties and inertness. As a result of these physical and chemical properties, PFCs can have ..."

2.3.5 page 3, line 13 - "Other sources include ..." is this still talking about the low mass PFCs? Maybe "Other sources of PFCs include..."

Done.

2.3.6 page 10, line 16 - perhaps replace 'sources' by 'pieces' or 'lines', and leave the word 'sources' to describe emissions to the atmosphere. Same on line 19.

Done.

"Sources of evidence" on these two lines has been replaced by "pieces of evidence".

2.3.7 page 10, line 21 - "This is slightly less..." - what is? The 2017 value is slightly less?

The reviewer is correct. Please see our response to comment 1.4.23, where we have rephrased the sentence.

2.3.8 page 12, line 2 - specify whether this is the interhemispheric ratio of emissions or mixing ratio

This refers to the IH ratio of mixing ratio. This is now made explicit in the text.

Original sentence: "If the interhemispheric ratio for n-C4F10 and n-C5F12 were to be similar to that of c-C4F8, then the differences between the Cape Grim and Tacolneston data would have to be at least 0.01 ppt."

Revised sentence: "If the interhemispheric ratios of the mixing ratios of n-C4F10 and n-C5F12 were to be similar to that of c-C4F8, then the differences between the Cape Grim and Tacolneston data for these two PFCs would have to be at least 0.01 ppt. "

2.3.9 page 12, line 22 - "The trends are somewhat similar" - to what? Each other? Another PFC?

Original sentence: "The trends of i-C6F14 and n-C6F14 are somewhat similar. "

Rephrased to: "The trends of i-C6F14 and n-C6F14 are somewhat similar to each other."

2.3.10 page 12, line 23 - The fastest increase occurred in the mid to late 1990s.

Original sentence: "Its fastest increase in atmospheric abundance occurred in the 1990s and has since slowed down."

Revised sentence: "The fastest increase in atmospheric abundance occurred in the 1990s and has since slowed down."

2.3.11 page 13, line 4 - I would add the years, i.e. "Mixing ratios in Taiwan in 2015 and 2016 are again ..."

Done.

2.3.12 page 15, line 21 - change to "for n-C4F10 (Fig. 8) and n-C5F12 (emissions < 5.3x10ôÅĂĂ5 Gg/yr)." as only emissions for n-C4F10 are shown in Fig 8. I know this is specified in the caption, but simpler for the reader to also specify here.

Done.

2.3.13 page 16, line 4 - Add "Cape Grim" as follows: "However the Cape Grim data in this time period"

Done.

2.3.14 1) page 16, line 15 - "although within the uncertainties of the early data set this is not a significant difference" - isn't it the sparsity of the data (i.e. timing of the increase depends on one sample, around 1996) rather than the data uncertainties as indicated in the figure that mean we can't be conclusive about the timing of each increase? 2) Is there a clue to the timing (of the n-C6F14 increase, at least) from the Ivy measurements of the Cape Grim air archive, as these are more dense in time?

1) We agree with the referee. The scarcity of the data is certainly the main reason we cannot be conclusive about the timing of the emissions increase. We have therefore revised our statement in the text.

Original text: "In contrast with n-C6F14, emission rates for i-C6F14 are estimated to have started increasing in 1992 (rather than in 1994, as estimated for n-C6F14, although within the uncertainties of the early data set this is not a significant difference) and its initial rate of increase is not as fast. It reaches a maximum of 0.25±0.02 Gg yr-1 in 1996-1997, which is when n-C6F14 emissions reach their maximum values as well."

Revised text: "Emission rates for i-C6F14 are estimated to have started increasing in 1992. This is in contrast to 1994, which is what was estimated for the onset of significant n-C6F14 emissions. However, the measurement uncertainties and especially the sparsity of the early data set do not allow for any further conclusions on the exact timing of the onset of emission increases for both isomers. i-C6F14 emissions increase at a slower rate than those for the n-isomer and reach a maximum of 0.25±0.02 Gg yr-1 in 1996-1997, which is when n-C6F14 emissions reach their maximum values as well."

2) Since Ivy et al. measurements represent a mixture of the C6F14 isomers, they cannot provide unequivocal determination of the timing of the onset of the emission increases of the individual isomers. This is especially the case since our data allow for the possibility of a difference in the timing of onset of increased emissions for the two isomers.

2.4.15 page 16, line 16 - What does 'It' refer to? i-C6F14 emissions, presumably, but please specify.

i-C4F14 emissions. Done (see revised sentence as part of our response to comment 2.4.15, point 1).

2.4.16 page 16, line 29 - 'peak emission rates have increased' - please be clear what you are comparing here (n-C6F14 alone and the sum n-C6F14 + i-C6F14?)

Done. Revised: "Two, peak emission rates of the sum of the C6F14 isomers have increased from . . ."

2.4.17 page 16, line 33 – 1) what is being compared to EDGAR emissions here? Are

you comparing the black dashes with the red dots? 2) Do they agree exceedingly well?

1) We are indeed comparing the EDGAR emissions (black dashes) with the red dots (i.e. the sum of the emissions of the two isomers of C6F14. This is explained in the legend and caption of Fig. 9. Note that the colour for this curve has been changed in the revised version, as part of our response to comment 1.6.35).

2) Given that all other PFC emissions do not compare well with the EDGAR emissions at all (i.e. differ several orders of magnitude), the sum of the emissions of the C6F14 isomers does indeed compare extremely well in a relative sense to the EDGAR emissions from 1999 onwards. We have revised the statement (see below).

Original sentence: "Three, the sum of the estimated emission rates of both C6F14 isomers agrees exceedingly well with the EDGARv4.2-reported values from 1999 onwards."

Revised sentence: "Three, the estimated emission rates of the sum of both C6F14 isomers agree exceedingly well (relative to all other PFCs reported here) with the EDGARv4.2-reported values from 1999 onwards."

2.4.18 page 18, line 1 and Table S4 - is this talking about the correlation with CO described at the end of page 9? Or just footprints? Please explain what this is.

The correlations referred to here in this sentence are those between PFC mixing ratios and the particle density footprints. The correlations with CO mixing ratios are only used to identify possible source types of PFCs measured in Taiwan. We have clarified this in the following way:

Original sentence(page 18, line 1): "Generally, the PFCs correlate less with any of the 15 regions used to quantify sources of particle densities used in the dispersion modelling, although sample numbers limit the statistics (Table S4)."

Revised sentence: "When PFC mixing ratios are correlated to the 15 regions used to quantify sources of particle densities used in the dispersion modelling, the R-squared

values are generally much smaller, although sample numbers limit the statistics (Table S4)."

Original sentence (page 18, line 10): "All PFCs correlate well with energy industry and domestic sources, which is linked to population density (Table S5)."

Revised sentence: "When using simulated CO mixing ratios as a tracer for source types, all PFCs correlate well with energy industry and domestic sources, which is linked to population density (Table S5)."

2.4.19 Fig 5 - I found this figure confusing because it is combined, and I think it would be less confusing if it were split into two panels. It is the use of different axes I found most confusing - it gives the wrong impression of the relative magnitude of the two isomers. I had to keep checking the caption/legend to remember which axis to use, and which isomer was which, as I came back to the plot a number of times as I read through the paper. The gridlines (although faint) don't match the right axis. The one advantage of combining the plots is that it is easy to compare the timing of the increase around 1995 for the two isomers, but this could also be done with one plot above the other. If the plot is to remain combined, I suggest plotting the different isomers on the same y-axis. Alternatively, if the different axis ranges are to be maintained then color the axis tick labels and text blue and green to correspond to the colors of the symbols and shaded areas of each isomer, as in Fig 8. You could also put 'i-C6F14' in blue text near the i-C6F14 curve, and similar for n-C6F14 (green text). I think would still prefer it split into 2 panels.

We thank the reviewer for the helpful suggestions in improving this figure. We have implemented these, except for the following:

- We chose not to plot the two isomers on the same axis or to adjust the scale of the secondary y-axis, because the trend of i-C6F14 would not be visible anymore. We agree that splitting the plot would be an alternative solution. However, to minimise the space used by the figures and to allow for a direct comparison of the C6F14 isomer

trends, we decided to keep plotting the data for these two isomers in one graph.

- We did not add the names of the isomers in a text box next to the trends, because we instead changed the colours of the axes and axes labels to match the colours of the trend lines.

We have made further improvements to the figure, as has been outlined in our response to comment 1.5.34.

2.4.20 Fig 5 - as the i-C6F14 data only begin around 1987, should the figure only show the red line and blue band from then? Otherwise it gives the impression that mixing ratio is known before 1987, but it isn't here.

We agree with this suggestion and have adjusted both the trend and emission plots for i-C6F14.

2.5.21 Fig 7 - could color the right y-axis text and the EDGAR data as a color other than black, to emphasise that it is on the secondary axis.

The EDGAR data is already distinguishable with the different line style, which is also used for the secondary y-axis. We decided not to change the colour in order to keep all emissions plots consistent with each other. We believe that consistency also helps the reader to interpret the graphs. We chose not to use a different colour for the EDGAR data in all emission plots, because it would be confusing in subsequent plots where various colour schemes are already used.

2.5.22 Fig 9 - it is hard to distinguish the red and magenta dotted lines on my printed copy - I suggest using a more different color or line type.

We have made changes to the colour scheme and line styles in order to make it easier to distinguish between the different lines, also on printed, grayscale copies. Please see our response to comment 1.6.35.

2.5.23 Figures - it might be good to combine the mixing ratio and emissions figures for

each PFC, for example combine Figs 3 and 7 as two panels in the same figure, Figs 4 and 8 etc. I found I wanted to look at the mixing ratio plot when I was reading about the emissions and looking at the emissions plot, and combining them would remove the need for flicking back and forth. The text could still remain in the same order, just describe the top (mixing ratio) panels first and come back to the lower (emissions) panels.

Implementing this suggestion would result in mixing ratio trends being less easy to compare between species and would additionally mean that the reader would have to flick back and forth when reading the emission section. We have therefore decided to leave the figures as they are (apart from changes mentioned in responses to previous comments).

2.5.24 The measurements should be made available in a data archive or as download-able tables in the Supplement.

See our response to comment 1.5.30.

Technical corrections:

2.5.25 page 12, line 18 – Hemisohere

Done.

2.5.26 page 12, line 19 - add 'is' after 'This'

Done.

2.5.27 page 16, line 26 - add 's' to 'emissions'

Done.

2.5.28 page 18, line 1 - with with

Done.

2.5.29 page 20, line 13 - exist, not exists

Done.

2.5.30 Supplement page 2, line 3 - Section ??

Done.
* * *
Responses to Referee #3

General Comments:

NA

Specific Comments:

3.1.1 P6 LN 14-16: Since the update of the calibration scale was suggested as one of the key purposes for this article, the absolute calibration procedure needs to be overviewed more in detail than in the current version, and thus readers don't have to search for and read through previous studies cited here.

We understand the concern of the reviewer. However, the calibration methods and procedures are exactly the same as described for the work of Laube et al. (2010). Additionally, if we were to include the absolute calibration procedure into our publication, then it would be put into the Supplement and readers would have to be referred to that too. Since the absolute calibration procedure is thoroughly described in detail in Laube et al.'s (2010) Supplement, the readers are referred to their work, as done in the revised text:

Original text: "The absolute calibration method has been described in detail in Laube et al. (2010) and has been described specifically in Oram et al. (2012) for c-C4F8 and in Laube et al. (2012) for C4F10, C5F12, C6F14 and C7F16, including linearity of the detector response and identification."

Revised text: "The absolute calibration method has been described in detail in Laube

et al. (2010) and has been described specifically in Oram et al. (2012) for c-C4F8 and in Laube et al. (2012) for C4F10, C5F12, C6F14, and C7F16, including linearity of the detector response and identification. For a detailed description and assessment of the calibration procedure and system we refer the reader to the Supplement of Laube et al. (2010). The only procedural difference for the calibrations here is the temperature of the system (100 °C instead of 80 °C)."

3.2.2 P 7, LN 8: Clarify that the combined influence is <2.8%. Is it a relative difference between the n-C4F10 background concentrations determined based on the two different scales?

We explicitly state that the combined influence of the i-isomer and the leak-tightness of the calibration system is less than 2.8% (page 7, line 7-8). In the subsequent sentence, we have rephrased our explanation of how we arrived at this number and in which we refer to added details in the Supplement:

Original sentence: "This has been determined by comparing the old and the new calibration scale for n-C4F10."

Revised sentence: "This value has been determined by calculating the relative difference of the n-C4F10 mixing ratio in the working standard using the two different (UEA2010 versus UEA2018) calibration scales (see more details in the Supplement)."

3.2.3 P 7, LN 14: What does the ion ratio means? Is it a ratio of peak heights for two chromatograms of m/z=219 and 169? Or a ratio of peak areas?

The ion ratio refers to the ratio of peak areas determined for ions with m/z = 219 and m/z = 169. This has been clarified in the text as so:

Original text: "The confidence in the complete separation of all isomers for C6F14 and C7F16 is based on the ratio of the two most abundant ions into which these compounds are ionised during mass spectrometry: (mass-to-charge (m/z) ratios 219.0 and 169.0)."

Revised text: "The confidence in the complete separation of all isomers for C6F14 and

C7F16 is based on the ratio of the peak areas of the two most abundant ions into which these compounds are ionised during mass spectrometry: (mass-to-charge (m/z) ratios 219.0 and 169.0)."

3.2.4 P 7, LN 14: Fig. S2 was shown earlier than Fig. S1 in the text.

Corrected.

3.2.5 P 7, LN 25: the accuracy derived from the CFC-11 reference compound seems to imply only an accuracy of the dilution process, not "the accuracy of the calibrations" that stated in the text. Otherwise more clarification is needed.

We have clarified that what we originally referred to as the "accuracy" is actually the offset between the CFC-11 calibration and NOAA values, which we use as a measure of accuracy (see our response to comment 1.3.18). We disagree with the referee that our measure of accuracy only represents the accuracy of the dilution process, as it also includes the accuracy of the entire pre-concentration and measurement process.

3.2.6 P 7, LN 30: How was the uncertainty of 0.3% determined? Was it 1-sigma? Then how many data were analyzed (n=?)?

It is indeed 1-sigma. N = 3. Please see our response to comment 1.3.17, in which we explain how this information has been incorporated into the text.

3.2.7 P 7, LN 31: How could the ranges (4.0-7.8%, and -5.5-2.8%) of calibration accuracy be determined? More detailed explanation should be given.

We have addressed this in our response to comment 1.3.17. Comment 1.5.31 enquired about the decision to report the range instead of the average. We decided to keep the range in brackets and add the average value.

3.2.8 P 7, LN 34: Again, please describe how the overestimation by 20% and 11% could be determined?

Please see our response to comment 1.3.20 by referee #1.

3.2.9 P 8, LN 15-19: The emission distribution for all PFCs seems to be based on the global distribution of C6F14 emissions recorded in the EDGAR database. The figures 7-10 showed the individual EDGAR emission estimate for each PFCs. Then authors need to discuss about how different (or consistent) the emission modeling results would be if the EDGAR emission estimate for each compound were used, instead of the C6F14 emissions record.

To clarify: the input to the model are the annual global emissions. The output is the mixing ratio per atmospheric layer per zonal band. We determine the annual global emissions per compound by iteratively changing the emission input depending on how well the model subsequently simulates the mixing ratios compared to the mixing ratios observed at Cape Grim. This is explained in Section 2.4 and references therein.

If the EDGAR emission estimates were used as input to the model, as is suggested in this comment, then the result is simply that the simulated mixing ratios will be a lot lower than what we simulate with our emission results, because the EDGAR emissions are a lot lower (except for C6F14).

However, we think that the referee meant us to discuss what the emission modelling results would be if we were to use the emission distribution of the EDGAR record for each PFC instead of the emission distribution of C6F14, which we opted to use and apply to all PFC runs.

This is indeed an interesting discussion, because there is a lot about the actual global emission distribution that we do not know. We did not strictly test using each PFC's EDGAR distribution in the model runs, because of the obvious incompleteness of the EDGAR database. For example, in the EDGAR database, all emission sources for n-C5F10 and C7F16 came exclusively from Romania and Japan, respectively. However, we did test a number of different emission distributions to see what the effect would be on the simulated mixing ratios. These emission distributions had in common that 99% of the emissions occurred in the Northern Hemisphere, but they differed in the fraction

of global emissions per latitudinal band. The conclusion from these runs is that, given the same global annual emissions, the simulated mixing ratios in the latitudinal band for Cape Grim (SH) are not affected by these variations in distribution. In contrast, the simulated mixing ratios for the latitudinal band for Tacolneston is affected due to the proximity of this latitudinal band to the emission sources.

Thus, even though we did not directly test the suggestion in this comment by the referee, we can say that if the emission distribution based on the EDGAR database for any PFC besides $C6F14$ is used, then the simulations for the mixing ratio at Cape Grim (which is used to compare to the observations at Cape Grim to obtain the global emission estimate) would not be significantly different compared to using the $C6F14$ emission distribution, but the simulated mixing ratios at Tacolneston would be impacted. Nevertheless, we feel that this topic has sufficiently been discussed on page 11, lines 5 to 17. Thus, we have decided not to include any further elaboration.

3.2.10 P 9, LN 6-8: This statement conflicts with a previous comment in page 7 (lines 8-10): "Due to the leak-tightness of the system and a lack of observed isomers, a revision of the calibration scales for c-C4F8 and n-C5F12 was deemed unnecessary. If within the calibration uncertainty, their re-calibration was not necessary, why an additional uncertainty besides the calibration uncertainty should be added for these compounds?

We do not agree that this is a conflicting statement. We justify that a re-calibration of c-C4F8 and n-C5F12 is unnecessary. The additional uncertainty that we attribute to these two particular compounds is not a calibration uncertainty, but an uncertainty in the determination of the mixing ratio in our current internal working standard. Each internal working standard has a different concentration. C-C4F8 and n-C5F12 had been calibrated for an internal working standard that we were using previously, so in order to obtain the mixing ratio of these PFCs in our current internal working standard, we converted the mixing ratio of the former to the latter by using the ratio in the signal of both internal working standards. This ratio carries an uncertainty, which is the uncertainty in question.

3.3.11 P 10, LN 6-11: Explain more in detail how a certain correlation between modeled CO and the PFC mixing ratio can represent which emission source is more associated with PFCs and what extent as well?

We have substituted line 31 on page 9 to line 11 on page 10 by the following:

"For the purpose of identifying the possible sources of PFCs, the NAME footprints were used to calculate the mixing ratio of a tracer at the Taiwan measurement site, given emissions from a range of different emission sectors. Carbon monoxide (CO) was chosen as the tracer because it is emitted from a number of different emission sources and because widely used and tested bottom-up emission estimates of CO are available from the Representative Concentration Pathway 8.5 (RCP 8.5) inventory (Riahi et al., 2011; Van Vuuren et al., 2011) (http://tntcat.iiasa.ac.at:8787/RcpDb/dsd?Action=htmlpage&page=welcome). Moreover, the RCP8.5 CO emissions are divided into various emission sectors (industry, power plants, solvents, agricultural waste burning, waste, forest burning, grassland burning, residential, international shipping, surface transportation, and agriculture), allowing us to test whether there was a correlation between the observed PFC mixing ratios and whether the air modelled to have arrived at the measurement site had likely been subject to emissions from any particular sector.

The NAME footprints were combined individually with the distribution of CO emissions from each sector to calculate a modelled mixing ratio of the emitted species at the measurement site (see Supplementary Material Section S5). Emissions were taken for the year 2010, for the timescale of each footprint (12 days) (more details can be found in Oram et al. (2017). A correlation analysis was performed between the modelled CO from each sector and the measured PFC mixing ratios. Industry (combustion and processing) and solvent applications are expected to show some correlations with PFC mixing ratios, as they are most closely associated with PFC sources and so we would expect their emissions to be co-located. Results of these analyses are found in Section 3.3."

We have also included an explanation about how the CO emissions per economic sector are calculated in the Supplementary Material (new Section S5).

3.3.12 P 10, LN 28: Clarify where the number of 1.05 came from.

Done.

Original sentence: ". . . , equivalent to an interhemispheric ratio of 1.05."

Revised sentence: ". . . , equivalent to an interhemispheric ratio of 1.05 (average enhancement of Tacolneston mixing ratio measurements against the background trend at Cape Grim)."

3.3.13 P 10, LN 29-32: Please compare the current c-C4F8 measurements in Tacolneston with those from Mace Head in Muhle et al. (2019) and discuss and justify the statement of "the lack of variability in the data suggests that Tacolneston is not in close proximity to any major sources of c-C4F8. When considering the simulated atmospheric trend from the emission model for Tacolneston, it can be noted that it overestimates the observed concentrations, despite the well-fitted Southern Hemispheric trend."

We have already included a comparison of the c-C4F8 Cape Grim time series to the Southern Hemisphere results in Muhle et al. (2019). From this comparison, it was concluded that there is no simple conversion factor between the two data sets. In turn, that means that we cannot use our Tacolneston and Muhle et al.'s Mace Head data to make any further statements on regional emissions. Please see our response to comment 1.4.23.

In addition, we have also modified our statement in lines 29-32 on page 10:

Original sentence: ". . . the lack of variability in the data suggests that Tacolneston is not in close proximity to any major sources of c-C4F8. When considering the simulated atmospheric trend from the emission model for Tacolneston, it can be noted that it overestimates the observed concentrations, despite the well-fitted Southern Hemi-

spheric trend."

Revised sentence: ". . . the lack of variability in the data suggests that Tacolneston is not in close proximity to any major sources of c-C4F8. Such a proximity of sources should cause multiple occurrences of substantially higher mixing ratios, as can be seen in the Taiwan data (discussed below). When considering the simulated atmospheric trend from the emission model for Tacolneston, it can be noted that it overestimates the observed concentrations, despite the well-fitted Southern Hemispheric trend. "

3.3.14 P 11, LN 6-8: Are they based on the current simulation? Otherwise, add the correspond references.

They are indeed based on the current simulation and therefore no additional reference is necessary.

3.3.15 P 11, LN 30-31: Please discuss how authors could exclude a possibility that the found similarity in the mixing ratio trends between n-C4F10 and n-C5F12 might be due to the fact that they were determined using a same m/z (119).

n-C4F10 and n-C5F12 elute at different times and therefore enter the mass spectrometer at a different time. Specifically, the retention time (i.e. the time after injection when the maximum of the signal peak appears on the chromatogram) for n-C4F10 on ion m/z 119 is 14.04 minutes, while for n-C5F12 on ion m/z 119 it is at 17.62 minutes. This is what distinguishes their peaks from each other, even though they're analysed on the same m/z ion. We do not feel that it is necessary to include this information in the paper.

3.3.16 P 112, LN 18: Much lower mixing ratio elevations were observed where? In Taiwan.

Original sentence: "Much lower mixing ratio elevations for both n-C4F10 and n-C5F12 were observed in 2015."

Sentence rephrased: "Much lower mixing ratio elevations for both n-C4F10 and n-

C5F12 were observed in Taiwan in 2015 compared to 2014 and 2016."

3.3.17 P 12, LN 33 – P 13, LN 2: 1) Please explain how author can confirm the ratio shift from 2003-2008 to 2013-2018 periods was statistically significant. How was the uncertainty of +/-0.01 determined? 2) Are there any possibility that the ratio shift could be related with the re-calibration after 2010?

1) We are giving the 1 sigma standard deviations of the ratios for these periods. This has been clarified in the text by adding it in brackets.

2) All samples have been re-measured (as described in Section 2.2), so this is not a possibility.

3.3.18 P 13, LN 10-11: For the observed high mixing ratio, authors need to examine their trajectories to argue an influence of a pollution plume.

Only two data points differ from the simulated Tacolneston by more than the 1-sigma measurement uncertainty. Both of these data points would agree within 2-sigma. We have therefore modified the statement as follows:

Original text: "Mixing ratios occasionally exceed the modelled uncertainty envelopes for Tacolneston, such as in 2015. This might imply a plume at the time of sampling. The simulated trend for the Tacolneston data corresponds well with the lower end of the variability in the observations."

Revised text: "Mixing ratios occasionally exceed the modelled uncertainty envelopes for Tacolneston, such as in 2015, but none of these excursions exceeded the 2-sigma measurement uncertainties. The simulated trend for the Tacolneston data corresponds well with the lower end of the variability in the observations."

3.4.19 P 18, LN 1: remove the first "with".

Done.

3.4.20 P 18, LN 10-17: Authors should describe much more in detail about the correlation analysis between PFCs mixing ratio versus CO mixing ratio derived CO source type, which was not given even in the Supplementary information. How can the CO mixing ratio be distinguished by the sources? The description and resulting discussion should be provided in the main text.

We have addressed this in our response to comment 3.3.11.

Please also note the supplement to this comment:
https://www.atmos-chem-phys-discuss.net/acp-2019-873/acp-2019-873-AC1-supplement.pdf
* * *
[Figure]

[Figure]

2016-04-29 02:30:00

0-100 m time-integrated particle density (g s m$^{-3}$)

**Fig. 1.**

[Figure]

**Fig. 2.**

Fig. 3.

**Fig. 4.**

[Figure]

**Fig. 5.**

[Figure]

Fig. 6.

[Figure]

**Fig. 7.**

[Figure]

- n-C$_4$F$_{10}$ modelled
- C$_4$F$_{10}$ modelled Laube et al.
- C$_4$F$_{10}$ modelled Ivy et al.
- C$_4$F$_{10}$ EDGAR
- n-C$_5$F$_{12}$ modelled
- C$_5$F$_{12}$ modelled Laube et al.
- C$_5$F$_{12}$ modelled Ivy et al.

**Fig. 8.**

Fig. 9.

[Figure]

**Fig. 10.**

Fig. 11.

Legend:
- i-C$_6$F$_{14}$
- n-C$_7$F$_{16}$
- n-C$_5$F$_{12}$
- n-C$_4$F$_{10}$
- n-C$_6$F$_{14}$
- c-C$_4$F$_8$

Y-axis: Million Metric Tonnes CO$_2$ Equivalent

X-axis: Year